# ReLU Regression with Massart Noise

**Ilias Diakonikolas**
University of Wisconsin-Madison
ilias@cs.wisc.edu

**Jongho Park**
University of Wisconsin-Madison
jongho.park@wisc.edu

**Christos Tzamos**
University of Wisconsin-Madison
tzamos@wisc.edu

## Abstract

We study the fundamental problem of ReLU regression, where the goal is to fit Rectified Linear Units (ReLUs) to data. This supervised learning task is efficiently solvable in the realizable setting, but is known to be computationally hard with adversarial label noise. In this work, we focus on ReLU regression in the Massart noise model, a natural and well-studied semi-random noise model. In this model, the label of every point is generated according to a function in the class, but an adversary is allowed to change this value arbitrarily with some probability, which is *at most $\eta < 1/2$*. We develop an efficient algorithm that achieves exact parameter recovery in this model under mild anti-concentration assumptions on the underlying distribution. Such assumptions are necessary for exact recovery to be information-theoretically possible. We demonstrate that our algorithm significantly outperforms naive applications of $\ell_1$ and $\ell_2$ regression on both synthetic and real data.

## 1 Introduction

Learning in the presence of outliers is a key challenge in machine learning, with several data analysis applications, including in ML security (4; 7; 46; 14) and in exploratory data analysis of real datasets with natural outliers, e.g., in biology (43; 41; 36). The goal is to design computationally efficient learners that can tolerate a constant fraction of outliers, independent of the dimensionality of the data. Early work in robust statistics (28; 30) gave sample-efficient robust estimators for various basic tasks, alas with exponential runtime. A recent line of work in computer science, starting with (13; 35), developed the first computationally efficient robust learning algorithms for various high-dimensional tasks. Since these early works, there has been significant progress in algorithmic robust high-dimensional statistics by several communities, see (16) for a recent survey.

In this work, we study the problem of learning Rectified Linear Units (ReLUs) in the presence of label noise. The ReLU function $\mathrm{ReLU}_{\mathbf{w}} : \mathbb{R}^d \to \mathbb{R}$, parameterized by a vector $\mathbf{w} \in \mathbb{R}^d$, is defined as $\mathrm{ReLU}_{\mathbf{w}}(\mathbf{x}) := \mathrm{ReLU}(\mathbf{w} \cdot \mathbf{x}) = \max\{0, \mathbf{w} \cdot \mathbf{x}\}$. ReLU regression – the task of fitting ReLUs to a set of labeled examples – is a fundamental task and an important primitive in the theory of deep learning. In recent years, ReLU regression has been extensively studied in theoretical machine learning both from the perspective of designing efficient algorithms and from the perspective of computational hardness, see, e.g., (26; 45; 37; 48; 27; 49; 11; 15; 25; 18). The computational difficulty of this problem crucially depends on the assumptions about the input data. In the realizable case, i.e., when the labels are consistent with the target function, the problem is efficiently solvable, see, e.g., (45). On the other hand, in the presence of even a small constant fraction of adversarially labeled data, computational hardness results are known even for approximate recovery (29; 37) and under well-behaved distributions (27; 15; 25; 18). See Section 1.3 for a detailed summary of related work.

35th Conference on Neural Information Processing Systems (NeurIPS 2021).

A challenging corruption model is the adversarial label noise model, in which an adversary is allowed to corrupt an *arbitrary* $\eta < 1/2$ fraction of the labels. The aforementioned hardness results rule out the existence of efficient algorithms for learning ReLUs with optimal error guarantees in this model, even when the underlying distribution on examples is Gaussian. Moreover, when no assumptions are made on the underlying data distribution, no fully polynomial time algorithm with non-trivial guarantee is possible. In fact, for the distribution-independent setting, even for the simpler case of learning linear functions, there are strong computational hardness results for any constant $\eta$ (33; 29). These negative results motivate the following natural question:

> *Are there realistic label noise models in which efficient learning is possible*
> *without strong distributional assumptions?*

Here we focus on ReLU regression in the presence of *Massart (or bounded) noise* (38), and provide an efficient learning algorithm with minimal distributional assumptions. In the process, we also provide an efficient noise-tolerant algorithm for the simpler case of linear regression.

The Massart model (38) is a classical semi-random noise model originally defined in the context of binary classification. In this model, an adversary has control over a *random* $\eta < 1/2$ fraction of the labels. Recent work (12) gave the first efficient learning algorithm for linear separators with non-trivial error gurrantees in the Massart model *without distributional assumptions*. In this work, we ask to what extent such algorithmic results are possible for learning real-valued functions. To state our contributions, we formally define the following natural generalization of the model for real-valued functions.

**Definition 1.1** (Learning Real-valued Functions with Massart Noise). *Let $\mathcal{F}$ be a concept class of real-valued functions over $\mathbb{R}^d$ and $f : \mathbb{R}^d \to \mathbb{R}$ be an unknown function in $\mathcal{F}$. For a given parameter $\eta < 1/2$, the algorithm specifies $m \in \mathbb{Z}_+$ and obtains $m$ samples $(\mathbf{x}_i, y_i)_{i=1}^m$, such that:*

- *(a) every $\mathbf{x}_i$ is drawn i.i.d. from a fixed distribution $\mathcal{D}_\mathbf{x}$, and*

- *(b) each $y_i$ is equal to $f(\mathbf{x}_i)$ with probability $1 - \eta$ and takes an* arbitrary *value with probability $\eta$, chosen by an adversary after observing the samples drawn and the values that can be corrupted.*

In the context of binary classification, the above model has been extensively studied in the theoretical ML community for the class of linear separators (2; 3; 51; 50; 20; 12; 9). Even though the noise model might appear innocuous at first sight, the ability of the Massart adversary to choose *whether* to perturb a given label and, if so, with what probability (which is unknown to the learner), makes the design of efficient algorithms in this model challenging. Specifically, for distribution-independent PAC learning of linear separators, even approximate learning in this model is computationally hard (17).

Extending this model to real-valued functions, we study regression under Massart noise for the realizable setting; that is, when the *uncorrupted* data exhibit clean functional dependencies, i.e., $y_i = \text{ReLU}(\mathbf{w}^* \cdot \mathbf{x}_i)$. The realizable setting is both of theoretical and practical interest. Prior work (45; 23; 31; 49) developed algorithms for learning ReLUs in this setting (without Massart noise), providing theoretical insights on the success of deep learning architectures. On the practical side, there are many applications in which we observe clean functional dependencies on the uncorrupted data. For instance, clean measurements are prevalent in many signal processing applications, including medical imaging, and are at the heart of the widely popular field of compressive sensing (8).

## 1.1 Main Results

To build up to the more challenging case of ReLUs, we start with the simpler case of linear functions, which is in and of itself one of the most well-studied statistical tasks, with numerous applications in machine learning (44), as well as in other disciplines, including economics (21) and biology (39).

In our Massart setting, the goal is to identify a linear relation $y = \mathbf{w}^* \cdot \mathbf{x}$ that the clean samples $(\mathbf{x}, y)$ (inliers) satisfy. We show that, under the minimal (necessary) assumption that the distribution is not fully concentrated on any subspace, the problem is efficiently identifiable.

**Theorem 1.2** (Exact Recovery of Linear Functions). *Let $\mathcal{D}_\mathbf{x}$ be a distribution on $\mathbb{R}^d$ that has zero measure on any linear subspace and let $\eta < 1/2$ be the upper bound on the Massart noise rate. Denote $\mathbf{w}^*$ the vector representing the true linear function. There is an algorithm that draws*

$\tilde{O}(\frac{d^3}{(1-2\eta)^2})$ *samples, runs in* $\text{poly}(d, b, (1 - 2\eta)^{-1})$ *time, where $b$ is an upper bound on the bit complexity of the samples and parameters, and outputs $\mathbf{w}^*$ with probability at least* $9/10$.

We provide a more general algorithmic result that relaxes the density assumption on $\mathcal{D}_{\mathbf{x}}$ in Theorem C.3 of Appendix C, so that the only assumption needed is that the support of $\mathcal{D}_{\mathbf{x}}$ spans $\mathbb{R}^d$. It is important to note that if the distribution was concentrated entirely on a linear subspace, it would be information-theoretically impossible to determine the orthogonal component of $\mathbf{w}^*$ on that subspace. That is, our anti-concentration assumption on the distribution is necessary for identifiability to be possible. Even when this assumption is violated and the problem is non-identifiable, we provide a (weaker) PAC learning guarantee for the linear case in Theorem D.1 of Appendix D.

Our main algorithmic result is for the problem of ReLU regression, where the inliers satisfy $y = \text{ReLU}(\mathbf{w}^* \cdot \mathbf{x})$ and an $\eta < 1/2$ fraction of the labels are corrupted by Massart noise. Even in this more challenging case, we show it is possible to efficiently identify the true parameters $\mathbf{w}^*$, as long as every homogeneous halfspace contains a non-negligible fraction of the sample points.

**Theorem 1.3** (Exact Recovery of ReLUs). *Let $\mathcal{D}_{\mathbf{x}}$ be a distribution on $\mathbb{R}^d$ that has zero measure on any linear subspace and such that $\Pr_{\mathbf{x} \sim \mathcal{D}_{\mathbf{x}}}[\mathbf{w} \cdot \mathbf{x} \geq 0] \geq \lambda$ for any $\mathbf{w} \in \mathbb{R}^d$. Let $\eta < 1/2$ be the upper bound on the Massart noise rate. Denote $\mathbf{w}^*$ the parameter vector of the target ReLU. There is an algorithm that draws $\tilde{O}(\frac{d^3}{\lambda^2(1-2\eta)^2})$ samples, runs in time $\text{poly}(d, b, \lambda^{-1}, (1 - 2\eta)^{-1})$, and outputs $\mathbf{w}^*$ with probability at least* $9/10$.

We similarly provide a more general result relaxing the density assumption on $\mathcal{D}_{\mathbf{x}}$ in Theorem C.4 of Appendix C. We note that the assumption on the mass of halfspaces in Theorem 1.3 is necessary for identifiability. Indeed, if there was a halfspace, parameterized by $\mathbf{w} \in \mathbb{R}^d$, such that $\Pr_{\mathbf{x} \sim \mathcal{D}_{\mathbf{x}}}[\mathbf{w} \cdot \mathbf{x} \geq 0] = 0$, it would be impossible to distinguish between the functions $\text{ReLU}(\mathbf{w} \cdot \mathbf{x})$ and $\text{ReLU}(2\mathbf{w} \cdot \mathbf{x})$ (even without noise), as all points would have $0$ labels. It remains an interesting open problem whether similar PAC learning guarantees can be obtained for the case of ReLU regression. We suspect that this problem is computationally hard in full generality.

## 1.2 Technical Overview

As explained in the introduction, the focus of our work is on the problem of robust regression in the presence of outliers. When $\eta < \frac{1}{2}$ fraction of the data is arbitrarily corrupted in the realizable setting, the goal is to compute the function that fits as many as points (inliers) as possible. Given a sufficient number of samples from a full-dimensional distribution, this function is unique for the class of ReLUs and matches the true function with high probability. However, even in the simpler case of linear functions, the corresponding computational problem of $\ell_0$-minimization is computationally hard without distributional assumptions, as it is an instance of robust subspace recovery (29).

Our positive results are driven by relaxing the assumption that an *arbitrary* $\eta$ fraction of the points is corrupted. Instead, as defined in Definition 1.1, we consider a more restricted adversary that is presented with a uniformly random $\eta$ fraction of the points, which can be corrupted arbitrarily at will.

$\ell_0$ **to** $\ell_1$ **minimization** Given this milder corruption model, we propose novel algorithms for efficient exact recovery of the underlying function. We obtain our algorithms by replacing the $\ell_0$-minimization with $\ell_1$-minimization, which can be shown to converge to the true function *in the limit* and is efficient to optimize in the linear regression case. For intuition, consider a single-point distribution that always outputs labeled examples of the form $(\mathbf{x}, y)$, where the example $\mathbf{x}$ is always the same but the labels $y$ may differ. The Massart assumption indicates that the value of $y$ is correct more than half of the time, so the estimate that maximizes the number of correct samples ($\ell_0$-minimizer) recovers the underlying function. However, if one considers the $\ell_1$-minimizer, i.e., the value $v$ that minimizes $\mathbb{E}[|y - v|]$, this corresponds to the median value of $y$ which is also correct if more than half samples are correct.

We generalize this intuition and propose a natural and tight condition under which empirical $\ell_1$-minimization results in the true $\ell_0$-minimizer (see Lemma 2.2). While this condition holds under Massart noise for arbitrary distributions in the population level, it can fail to hold with high probability when considering only a finite set of samples from the distribution. For example, consider the one-dimensional case of $\mathbf{w}^* = 1$ where most $\mathbf{x}_i$'s are near-zero and uncorrupted, while a few corrupted samples lie extremely far from zero. The empirical $\ell_1$-minimizer here will be dominated by the few

corrupted samples and would not be the $\ell_0$-minimizer. In particular, the sample complexity of naive $\ell_1$-minimization would crucially depend on the concentration properties of the distribution on $\mathbf{x}$.

**Transforming the points via radial-isotropy** The main technical idea behind obtaining sample-efficient algorithms that run in polynomial time is to transform the original point set into an equivalent one that satisfies the required properties with high probability, as it becomes sufficiently concentrated. In particular, performing a linear transformation mapping every point $\mathbf{x}$ to $\mathbf{Ax}$, while keeping the corresponding label $y$, is without loss of generality, as we are interested in identifying the true (generalized) linear function that depends only on the inner product of every point with a parameter vector $\mathbf{w}$. Finding such a vector $\mathbf{w}'$ in the transformed space $\mathbf{Ax}$ results in the equivalent vector $\mathbf{w} = \mathbf{A}^T\mathbf{w}'$ in the original space. Moreover, an additional operation we can perform is to take a single sample $(\mathbf{x}, y)$ and multiply it by a positive scalar $\lambda > 0$ to replace it with the sample $(\lambda\mathbf{x}, \lambda y)$. For both the linear and ReLU cases, any sample that is an inlier for the true function remains an inlier after this transformation.

We can use these two operations to bring our point set in radial-isotropic position, i.e., so that all the $\mathbf{x}$'s in the dataset are unit-norm and the variance in any direction is nearly identical:

**Definition 1.4** (Radial Isotropy). *Let $\mathcal{S}^{d-1}$ be the unit sphere in $\mathbb{R}^d$. Given $\{\mathbf{x}_1, \ldots, \mathbf{x}_n\} \subset \mathcal{S}^{d-1}$, $\mathbf{A} : \mathbb{R}^d \to \mathbb{R}^d$ is a radial-isotropic transformation if $\sum_{i=1}^n \frac{(\mathbf{Ax}_i)(\mathbf{Ax}_i)^T}{\|\mathbf{Ax}_i\|_2^2} = \frac{n}{d}I$. For $0 < \gamma < 1$, we say the points are in $\gamma$-approximate radial-isotropic position, if for all $\mathbf{v} \in \mathcal{S}^{d-1}$, it holds that $(d/n)\sum_{i=1}^n (\mathbf{x}_i \cdot \mathbf{v})^2 \geq 1 - \gamma$.*

In such a normalized position, we can argue that with high probability the weight of all inliers in every direction is more than the weight of the outliers, which guarantees that the empirical $\ell_1$-minimizer will converge to the true function.

**Learning ReLUs** Unfortunately, while $\ell_1$-minimization for linear functions is convex and efficiently solvable via linear programming, $\ell_1$-minimization for ReLUs is challenging due to its non-convexity; that is, we cannot easily reduce ReLU regression to a simple optimization method. We instead establish a structural condition under which we can compute an efficient separation oracle between the optimal parameter vector $\mathbf{w}^*$ and a query $\mathbf{w}$. More specifically, we show that any suboptimal guess for the parameter vector $\mathbf{w}$ can be improved by moving along the opposite direction of the gradient of the $\ell_1$-loss for the subset of points in which the condition in Lemma 3.1 is satisfied. Identifying such a direction of improvement yields a separating hyperplane, so we exploit this to efficiently identify $\mathbf{w}^*$ by running the ellipsoid method with our separation oracle.

However, for this result to hold with a small number of samples, we need to again bring to radial-isotropic position the points that fall in the linear (positive) part of the ReLU for the current guess vector $\mathbf{w}$. In contrast to the linear case, though, where this transformation was applied once, in this case it needs to be applied again with every new guess. This results in a function that changes at every step, which is not suitable for direct optimization.

Using these ideas, our algorithms can efficiently recover the underlying function exactly using few samples. Our algorithms make mild genericity assumptions about the position of the points, requiring that the points are not concentrated on a lower-dimensional subspace or, for the case of ReLUs, do not lie entirely in an origin-centered halfspace. As already mentioned, such assumptions are necessary for the purposes of identifiability.

## 1.3 Related Work

Given the extensive literature on robust regression, here we discuss the most relevant prior work.

**ReLU Regression** In the realizable setting, (45) and, more recently, (31) showed that gradient descent efficiently performs exact recovery for ReLU regression under the Gaussian distribution on examples. (49) generalized this result to a broader family of well-behaved distributions. In the agnostic or adversarial label noise model, a line of work has shown that learning with near-optimal error guarantees requires super-polynomial time, even under the Gaussian distribution (27; 15; 25; 18). On the positive side, (11) gave an efficient learner with approximation guarantees under log-concave distributions. Without distributional assumptions, even approximate learning is hard (29; 37).

The recent work (32) studies ReLU regression in the realizable setting under a noise model similar to – but more restrictive than – the Massart model of Definition 1.1. Specifically, in the setting of (32), the adversary can corrupt a label with probability at most $\eta$, but only via additive noise bounded above by a constant. (32) gives an SGD-type algorithm for ReLU regression in this model. We note that their algorithm does not achieve exact recovery and its guarantees crucially depend on the concentration properties of the marginal distribution and the bound on the additive noise.

**Comparison of Noise models**   It is worth comparing the Massart noise model (Definition 1.1) with other noise models studied in the literature. The strongest corruption model we are aware of is the strong contamination model (13), in which an omniscient adversary can corrupt an arbitrary $\eta < 1/2$ fraction of the labeled examples. In the adversarial label noise model, the adversary can corrupt an arbitrary $\eta < 1/2$ fraction of the labels (but not the examples). Efficient robust learning algorithms in these models typically only give approximate error guarantees and require strong distributional assumptions. Specifically, for the case of linear regression, (34; 19; 14) give robust approximate learners in the strong contamination model under the Gaussian distribution and, more broadly, distributions with bounded moments. In the adversarial label noise model, (6) gave efficient robust learners under strong concentration bounds on the underlying distribution that can tolerate $\eta < 1/50$ fraction of outliers.

The recent work (10) considers a Massart-like noise model in the context of linear regression with random observation noise. (10) provides an SDP-based approximate recovery algorithm when the noise rate satisfies $\eta < 1/3$. It should be noted their algorithm does not efficiently achieve exact recovery. Due to space limitations, we provide a more detailed description of that work in Appendix F.

A related noise model is that of *oblivious* label noise, where the adversary can corrupt an $\eta$ fraction of the labels with additive noise that is *independent* of the covariate $\mathbf{x}$. More precisely, the oblivious adversary corrupts the vector of labels $\mathbf{y} \in \mathbb{R}^m$ by adding a $\eta m$-sparse corruption vector $\mathbf{b}$. Since $\mathbf{b}$ is independent of the covariates, oblivious noise can be viewed as corrupting a sample with probability $\eta$ with a random non-zero entry of $\mathbf{b}$. Consequently, oblivious noise can be seen as a special case of Massart noise. We formally compare these two noise models in more detail in Appendix E. A line of work (5; 47; 22; 42) studied robust linear regression under oblivious noise and developed efficient exact recovery algorithms under strong distributional assumptions.

## 2   Warmup: Linear Regression with Massart Noise

In this section, we establish structural conditions under which we can perform efficient $\ell_0$-minimization for linear functions under Massart noise. It is imperative that we find the $\ell_0$-minimizer with respect to $\mathbf{w}$ since, with a sufficient number of samples, the $\ell_0$-minimizer is the true function we wish to recover. We then show that appropriately transforming the data via radial-isotropic transformation and then solving for the empirical $\ell_1$-loss $\arg\min_{\mathbf{w} \in \mathbb{R}^d} \frac{1}{m} \sum_{i=1}^{m} |\tilde{y}_i - \mathbf{w} \cdot \tilde{\mathbf{x}}_i|$ can efficiently recover the true parameter $\mathbf{w}^*$. We describe the algorithm for recovering linear functions below.

---

**Algorithm 1** Linear function recovery via radial isotropy

---

Draw $m = \tilde{O}(\frac{d^3}{(1-2\eta)^2})$ samples $(\mathbf{x}_i, y_i)_{i=1}^{m}$ with $\eta$-Massart noise

Compute $\mathbf{A}$ that puts $(\mathbf{x}_i, y_i)_{i=1}^{m}$ in $1/2$-approximate radial-isotropic position

$\hat{\mathbf{w}} \leftarrow \arg\min_{\mathbf{w} \in \mathbb{R}^d} \sum_{i=1}^{m} \left| \frac{y_i}{\|\mathbf{A}\mathbf{x}_i\|_2} - \mathbf{w} \cdot \frac{\mathbf{A}\mathbf{x}_i}{\|\mathbf{A}\mathbf{x}_i\|_2} \right|$ by solving the LP

**return** $\mathbf{A}\hat{\mathbf{w}}$

---

In fact, there is no need to compute an exact radial-isotropic transformation ($\gamma = 0$) as an approximate one suffices. An approximate radial-isotropic transformation can be computed efficiently as in Lemma 2.1, which we prove in Appendix A.

**Lemma 2.1.** *Given $S \subset \mathbb{R}^d$ in general position, there is a $\mathrm{poly}(n, d, b, \gamma^{-1})$ time algorithm that computes a positive definite symmetric matrix $\mathbf{A}$ such that $\left\{ \frac{\mathbf{A}\mathbf{x}}{\|\mathbf{A}\mathbf{x}\|} : \mathbf{x} \in S \right\}$ is in $\gamma$-approximate radial-isotropic position where $b$ is an upper bound on the bit complexity of the parameters and samples in $S$. Morever, the condition number of $\mathbf{A}$ is at most $2^{\mathrm{poly}(n,d,b)}$.*

Given that computing such approximate transformation $\mathbf{A}$ and solving a linear program (LP) can be done efficiently, Algorithm 1 gives us the polynomial runtime for Theorem 1.2.

The proof of Theorem 1.2 relies on two key ideas. First, we can minimize the empirical $\ell_1$-loss with respect to $\mathbf{w}$ instead of the $\ell_0$-loss because the two are equivalent under the structural condition of Lemma 2.2. Once the following sufficient condition is satisfied, we can efficiently solve for $\ell_1$-minimization as this is efficient in the case of linear functions by solving an LP.

**Lemma 2.2** (Structural Condition for Recovery). *Given $f : \mathbb{R} \to \mathbb{R}$ and $m$ samples $(\mathbf{x}_i, y_i)_{i=1}^m$ in $\mathbb{R}^d$, let the $\ell_0$-minimizer $\mathbf{w}^* = \arg\min_{\mathbf{w} \in \mathbb{R}^d} \frac{1}{m} \sum_{i=1}^m \|y_i - f(\mathbf{w} \cdot \mathbf{x}_i)\|_0$ be unique. If*

$$\sum_{y_i = f(\mathbf{w}^* \cdot \mathbf{x}_i)} |f((\mathbf{w}^* + \mathbf{r}) \cdot \mathbf{x}_i) - f(\mathbf{w}^* \cdot \mathbf{x}_i)| > \sum_{y_i \neq f(\mathbf{w}^* \cdot \mathbf{x}_i)} |f((\mathbf{w}^* + \mathbf{r}) \cdot \mathbf{x}_i) - f(\mathbf{w}^* \cdot \mathbf{x}_i)| \quad (\star)$$

*for all non-zero $\mathbf{r} \in \mathbb{R}^d$, then $\mathbf{w}^*$ is also the $\ell_1$-minimizer $\arg\min_{\mathbf{w} \in \mathbb{R}^d} \frac{1}{m} \sum_{i=1}^m |y_i - h(\mathbf{x}_i)|$.*

The structural condition $(\star)$ for linear functions reduces to having the sum of $|\mathbf{r} \cdot \mathbf{x}_i|$ for the "good" points be greater than the sum of $|\mathbf{r} \cdot \mathbf{x}_i|$ for the "bad" points in every direction $\mathbf{r}$. However, this implies that if one sample is much greater in norm than the others in some direction, this point can have undue influence and may easily dominate the $\ell_1$-loss. Therefore, without any preprocessing or transformation to the data, one has to rely on naively increasing the sample complexity until there are enough points in this direction to satisfy condition $(\star)$. Instead, we minimize the dominating effects of such outlier points and reduce the sample complexity through transforming the dataset with radial isotropy. We defer the proof of Lemma 2.2 and Theorem 1.2 to Appendix A.

## 3 ReLU Regression with Massart Noise

In this subsection, we study the problem of exact recovery for ReLUs in the presence of Massart noise. For the case of ReLUs, we can still use the structural condition of Lemma 2.2 to do $\ell_1$-minimization $\arg\min_{\mathbf{w}} \frac{1}{m} \sum_{i=1}^m |y_i - \mathrm{ReLU}(\mathbf{w} \cdot \mathbf{x}_i)|$. However, minimizing this objective is no longer straightforward, because the objective function is non-convex. Despite this fact, it is possible to exactly recover a ReLU under mild anti-concentration assumptions on the underlying distribution.

The key idea behind Theorem 1.3 is establishing the condition under which we can compute an efficient separation oracle between the query $\mathbf{w}$ and the true parameter $\mathbf{w}^*$. Once we obtain a separation oracle, we can use the ellipsoid method to recover $\mathbf{w}^*$ exactly. In turn, similarly to Lemma 2.2, we identify a sufficient structural condition on the dataset, which allows us to efficiently compute a separating hyperplane between $\mathbf{w}$ and $\mathbf{w}^*$ if $\mathbf{w} \neq \mathbf{w}^*$, and then use radial-isotropic transformations such that this condition is satisfied. We state this separation condition in the following lemma.

**Lemma 3.1** (Separation Condition). *Let $\mathcal{H}$ be an hypothesis class such that $\mathcal{H} = \{h_\mathbf{w} : h_\mathbf{w}(\mathbf{x}) = f(\mathbf{w} \cdot \mathbf{x}), \mathbf{w} \in \mathbb{R}^d\}$ where $f : \mathbb{R} \to \mathbb{R}$ is monotonically non-decreasing. Given a set of $m$ samples $(\mathbf{x}_i, y_i)_{i=1}^m$, let $\mathbf{w}^* = \arg\min_{\mathbf{w} \in \mathbb{R}^d} \frac{1}{m} \sum_{i=1}^m \|y_i - f(\mathbf{w} \cdot \mathbf{x}_i)\|_0$ be unique. Let $\Delta > 0$ and $\mathcal{B}(\mathbf{w}, \Delta)$ be the open ball of radius $\Delta$ centered at $\mathbf{w}$. Denote the empirical $\ell_1$-loss $\hat{L}(\mathbf{w}) = (1/m) \sum_{i=1}^m |y_i - f(\mathbf{w} \cdot \mathbf{x}_i)|$. Given a query $\mathbf{w}_0 \notin \mathcal{B}(\mathbf{w}^*, \Delta)$, if*

$$\sum_{y_i = f(\mathbf{w}^* \cdot \mathbf{x}_i)} |(\mathbf{w}_0 - \mathbf{w}^*) \cdot \mathbf{x}_i| f'(\mathbf{w}_0 \cdot \mathbf{x}_i) - \sum_{y_i \neq f(\mathbf{w}^* \cdot \mathbf{x}_i)} |(\mathbf{w}_0 - \mathbf{w}^*) \cdot \mathbf{x}_i| f'(\mathbf{w}_0 \cdot \mathbf{x}_i) \geq \Delta m \quad (\dagger)$$

*then $\nabla \hat{L}(\mathbf{w}_0) \cdot (\mathbf{w}_0 - \mathbf{w}) = 0$ is a separating hyperplane for $\mathbf{w}_0$ and $\mathcal{B}(\mathbf{w}^*, \Delta/2)$ such that $\nabla \hat{L}(\mathbf{w}_0) \cdot (\mathbf{w}_0 - \mathbf{w}) > 0$ for $\mathbf{w} \in \mathcal{B}(\mathbf{w}^*, \Delta/2)$.*

In particular, the gradient of the empirical $\ell_1$-loss gives us the separating hyperplane above. Other than the fact that only the points in the non-negative side of the halfspace of $\mathbf{w}$ are considered in the separation condition $(\dagger)$, the condition resembles much of the structural condition used for linear functions. Analogously, we apply a radial-isotropic transformation to the points of $\mathbf{w} \cdot \mathbf{x}_i \geq 0$. Thus, we have the following sub-procedure of the ellipsoid method where $\Delta$ is the radius of a ball which depends on the distance between the points (and hence the bit complexity $b$).

The main difference between the algorithm for ReLUs and linear functions is that here we must apply a different radial-isotropic transformation to every new subset of points in every iteration, depending on the query $\mathbf{w}_0$. In turn, the algorithm transforms the space according a new transformation $\mathbf{A}$ and computes a separating hyperplane and transforms the hyperplane back into the original space. Due to

---

**Algorithm 2** Separation oracle sub-procedure

---

**Input:** $(\mathbf{x}_i, y_i)_{i=1}^m$ with Massart noise, query $\mathbf{w}_0$, $\Delta > 0$
**Output:** Unless $\mathbf{w}_0 \in \mathcal{B}(\mathbf{w}^*, \Delta)$, a separating hyperplane between $\mathbf{w}_0$ and $\mathcal{B}(\mathbf{w}^*, \Delta/2)$
**if** $\mathrm{ReLU}(\mathbf{w}_0 \cdot \mathbf{x})$ fits at least $\frac{m}{2}$ points **then**
    **return** $\mathbf{w}_0$ as true parameter $\mathbf{w}^*$
$S \leftarrow \{(\mathbf{x}_i, y_i) : \mathbf{w}_0 \cdot \mathbf{x}_i \geq 0 \text{ for } i \in [m]\}$
Compute $\mathbf{A}$ that puts $S_{\mathbf{x}}$ into $1/2$-approximate radial-isotropic position
$\mathbf{r} \leftarrow \frac{1}{|S|} \sum_{(\mathbf{x}_i, y_i) \in S} \frac{\mathbf{A}\mathbf{x}_i}{\|\mathbf{A}\mathbf{x}_i\|_2} \cdot \mathrm{sgn}(\mathbf{w}_0 \cdot \mathbf{x}_i - y_i)$
**return** separating hyperplane $\mathbf{A}^{-1}\mathbf{r} \cdot (\mathbf{w}_0 - \mathbf{w}) = 0$

---

these repeated transformations, the proof of Theorem 1.3 requires a more intricate argument to make the ellipsoid method work correctly. We now prove Theorem 1.3.

*Proof of Theorem 1.3.* Let $\mathbf{w}_0$ be the original query to the oracle and assume the separation condition (†) holds for a set of points $(\mathbf{A}\mathbf{x}_i/\|\mathbf{A}\mathbf{x}_i\|_2, y_i/\|\mathbf{A}\mathbf{x}_i\|_2)_{i=1}^m$ and $\mathbf{A}^{-1}\mathbf{w}_0$. Then, by Lemma 3.1, $\mathbf{r} \cdot (\mathbf{A}^{-1}\mathbf{w}_0 - \mathbf{w}) = 0$ where $\mathbf{r} = (1/m) \sum_{i=1}^m (\mathbf{A}\mathbf{x}_i/\|\mathbf{A}\mathbf{x}_i\|_2) \cdot \mathrm{sgn}(\mathbf{w}_0 \cdot \mathbf{x}_i - y_i)$ separates $\mathbf{A}^{-1}\mathbf{w}_0$ and $\mathbf{A}^{-1}\mathbf{w}^*$. Thus the separation for $\mathbf{w}_0$ and $\mathbf{w}^*$ is $\mathbf{A}^{-1}\mathbf{r} \cdot (\mathbf{w}_0 - \mathbf{w}) = 0$.

Now it remains to check the sample complexity necessary for the separation condition (†). Each unique set of $\{(\mathbf{x}_i, y_i) : \mathbf{w}_0 \cdot \mathbf{x}_i \geq 0 \text{ for } i \in [m]\}$ determines a radial isotropic transformation but there can only be at most $m^{d+1}$ unique sets by the VC dimension of halfspaces. So there are only at most $m^{d+1}$ radial-isotropic transformations we have to consider. Let $\mathbf{A}$ be the linear transformation of the radial-isotropic transformation applied to points of $\mathbf{w}_0 \cdot \mathbf{x}_i \geq 0$. Denote $(\tilde{\mathbf{x}}_i, \tilde{y}_i) = (\mathbf{A}\mathbf{x}_i/\|\mathbf{A}\mathbf{x}_i\|_2, y_i/\|\mathbf{A}\mathbf{x}_i\|_2)$, $\tilde{\mathbf{w}}^* = \mathbf{A}^{-1}\mathbf{w}^*$, $\tilde{\mathbf{w}}_0 = \mathbf{A}^{-1}\mathbf{w}_0$, and let $\tilde{\mathcal{D}}_{\mathbf{x}}$ be $\mathcal{D}_{\mathbf{x}}|_{\{\mathbf{w}_0 \cdot \mathbf{x} \geq 0\}}$ transformed by $\mathbf{A}$ then normalized so that $\tilde{\mathcal{D}}_{\mathbf{x}}$ lies on $\mathcal{S}^{d-1}$. Then, for all $m^{d+1}$ transformations, we have the following VC inequality using $m = \tilde{O}(d/\epsilon^2)$ samples with high probability:

$$\sup_{\mathbf{w} \in \mathbb{R}^d} \Big| \Pr_{\tilde{\mathbf{x}} \sim \tilde{\mathcal{D}}_{\mathbf{x}}} \big[ |(\mathbf{w} - \tilde{\mathbf{w}}^*) \cdot \tilde{\mathbf{x}}| \mathbb{1}\{\mathbf{w} \cdot \tilde{\mathbf{x}} \geq 0, \ y = \mathrm{ReLU}(\mathbf{w}^* \cdot \mathbf{x})\} > t \big]$$

$$- (1/m) \sum_{i=1}^m \mathbb{1}\{|(\mathbf{w} - \tilde{\mathbf{w}}^*) \cdot \tilde{\mathbf{x}}_i| > t, \ \mathbf{w} \cdot \tilde{\mathbf{x}}_i \geq 0, \ y_i = \mathrm{ReLU}(\mathbf{w}^* \cdot \mathbf{x}_i)\} \Big| \leq \epsilon.$$

Let $S = \{(\tilde{\mathbf{x}}_i, \tilde{y}_i) : \mathbf{w}_0 \cdot \mathbf{x}_i \geq 0\}$. Similarly to the proof of Theorem 1.2, we have that $\max_{(\tilde{\mathbf{x}}, \tilde{y}) \in S} |\mathbf{r} \cdot \tilde{\mathbf{x}}| \leq \frac{(1-\gamma)d}{1-\gamma-\epsilon d} \mathbb{E}_{\tilde{\mathbf{x}} \sim \tilde{\mathcal{D}}_{\mathbf{x}}} [|\mathbf{r} \cdot \tilde{\mathbf{x}}|]$ where $\gamma = 1/2$. Then, we can write

$$(1/|S|) \sum_{(\tilde{\mathbf{x}}_i, \tilde{y}_i) \in S} |(\tilde{\mathbf{w}}_0 - \tilde{\mathbf{w}}^*) \cdot \tilde{\mathbf{x}}| \mathbb{1}\{y_i = \mathrm{ReLU}(\mathbf{w}^* \cdot \mathbf{x}_i)\}$$

$$= (m/|S|) \int_0^\infty \big( (1/m) \sum_{i=1}^m \mathbb{1}\{|(\tilde{\mathbf{w}}_0 - \tilde{\mathbf{w}}^*) \cdot \tilde{\mathbf{x}}_i| > t, \ \tilde{\mathbf{w}}_0 \cdot \tilde{\mathbf{x}}_i \geq 0, \ y_i = \mathrm{ReLU}(\mathbf{w}^* \cdot \mathbf{x}_i)\big) dt$$

$$\geq \mathbb{E}_{\tilde{\mathcal{D}}_{\mathbf{x}}} [|(\tilde{\mathbf{w}}_0 - \tilde{\mathbf{w}}^*) \cdot \tilde{\mathbf{x}}| \mathbb{1}\{y = \mathrm{ReLU}(\mathbf{w}^* \cdot \mathbf{x})\}] - (\epsilon m/|S|) \max_{(\tilde{\mathbf{x}}, \tilde{y}) \in S} |(\tilde{\mathbf{w}}_0 - \tilde{\mathbf{w}}^*) \cdot \tilde{\mathbf{x}}|$$

$$\geq (1-\eta) \mathbb{E}_{\tilde{\mathcal{D}}_{\mathbf{x}}} [|(\tilde{\mathbf{w}}_0 - \tilde{\mathbf{w}}^*) \cdot \tilde{\mathbf{x}}|] - (\epsilon m d/(|S|(1-2\epsilon d))) \mathbb{E}_{\tilde{\mathcal{D}}_{\mathbf{x}}} [|(\tilde{\mathbf{w}}_0 - \tilde{\mathbf{w}}^*) \cdot \tilde{\mathbf{x}}|]$$

By setting $\epsilon = \tilde{O}(\lambda(1 - 2\eta)/d)$, we can bound $m/|S|$ be at most a constant times $\lambda^{-1}$ for all $m^{d+1}$ possible subsets $S$ using Hoeffding's inequality and the union bound. Then we have

$$(1/|S|) \sum_{(\tilde{\mathbf{x}}, \tilde{y}) \in S} |(\tilde{\mathbf{w}}_0 - \tilde{\mathbf{w}}^*) \cdot \tilde{\mathbf{x}}| \mathbb{1}\{\tilde{y} = \mathrm{ReLU}(\tilde{\mathbf{w}}^* \cdot \tilde{\mathbf{x}})\} \geq ((1/2) + (1 - 2\eta)/4) \mathbb{E}_{\tilde{\mathcal{D}}_{\mathbf{x}}} [|(\tilde{\mathbf{w}}_0 - \tilde{\mathbf{w}}^*) \cdot \tilde{\mathbf{x}}|].$$

We can do the same to the corrupted points in $S$ getting $\leq (1/2 - (1 - 2\eta)/4) \mathbb{E}_{\tilde{\mathcal{D}}_{\mathbf{x}}} [|(\tilde{\mathbf{w}} - \tilde{\mathbf{w}}^*) \cdot \tilde{\mathbf{x}}|]$. Thus, for points $(\tilde{\mathbf{x}}, \tilde{y}) \in S$, we have the condition

$$(1/|S|) \Big( \sum_{\tilde{y} = \mathrm{ReLU}(\tilde{\mathbf{w}}^* \cdot \tilde{\mathbf{x}})} |(\tilde{\mathbf{w}}_0 - \tilde{\mathbf{w}}^*) \cdot \tilde{\mathbf{x}}| - \sum_{\tilde{y} \neq \mathrm{ReLU}(\tilde{\mathbf{w}}^* \cdot \tilde{\mathbf{x}})} |(\tilde{\mathbf{w}}_0 - \tilde{\mathbf{w}}^*) \cdot \tilde{\mathbf{x}}| \Big) \geq (1/2 - \eta) \mathbb{E}_{\tilde{\mathcal{D}}_{\mathbf{x}}} [|(\tilde{\mathbf{w}}_0 - \tilde{\mathbf{w}}^*) \cdot \tilde{\mathbf{x}}|]$$

$$\geq (1/2 - \eta) \|\tilde{\mathbf{w}}_0 - \tilde{\mathbf{w}}^*\|_2/d \ .$$

By Lemma 3.1, the inequality above implies that we can find a hyperplane of $\mathbf{r}$ that separates $\tilde{\mathbf{w}}_0$ and $\mathcal{B}(\tilde{\mathbf{w}}^*, \tilde{\Delta}/2)$ where $\tilde{\Delta} = \frac{1-2\eta}{2d}\|\tilde{\mathbf{w}}_0 - \tilde{\mathbf{w}}^*\|_2$. In the original space of $(\mathbf{x}_i, y_i)_{i=1}^m$, we have that the transformed hyperplane of $\mathbf{A}^{-1}\mathbf{r}$ seperates $\mathbf{w}_0$ and $\mathcal{B}(\mathbf{w}^*, \Delta/2)$ where $\Delta = \frac{1-2\eta}{2d} \cdot \frac{\lambda_{\min}(\mathbf{A})}{\lambda_{\max}(\mathbf{A})}\|\mathbf{w}_0 - \mathbf{w}^*\|_2$ since applying $\mathbf{A}^{-1}$ to $\mathbf{r}$ keeps the distance from $\mathbf{w}^*$ to $\mathbf{w}_0$ at least $\frac{1-2\eta}{2d\lambda_{\max}(\mathbf{A})}\|\mathbf{w}_0 - \mathbf{w}^*\|_2$ and applying $\mathbf{A}$ to $\tilde{\mathbf{w}}^*$ bounds the distance from $\mathbf{w}^*$ to $\mathbf{w}_0$ to be at least $\Delta$. Therefore, we can set $\Delta$ of Algorithm 2 to be equal to $\min_{\mathbf{w} \neq \mathbf{w}^*} \frac{1-2\eta}{2d} \cdot \frac{\lambda_{\min}(\mathbf{A})}{\lambda_{\max}(\mathbf{A})}\|\mathbf{w} - \mathbf{w}^*\|_2$.

If $\mathbf{w} \neq \mathbf{w}^*$, by bounded bit complexity $b$, we have that the volume of the ellipsoid decreases at every step but the ball of radius $\frac{(1-2\eta)\lambda_{\min}(\mathbf{A})}{4d\lambda_{\max}(\mathbf{A})}$ will always be contained in it. Thus the algorithm terminates in $\mathrm{poly}(d, b, (1-2\eta)^{-1})$ iterations since $\log\left(\frac{\lambda_{\max}(\mathbf{A})}{\lambda_{\min}(\mathbf{A})}\right) = \mathrm{poly}(d, b, (1-2\eta)^{-1})$. $\qquad\square$

## 4 Experiments

In this section, we apply our algorithms that are based on radial-isotropic transformations to both synthetic and real datasets and compare robustness in regression with other baseline methods of $\ell_1$ and $\ell_2$-regression. Our experiments demonstrate the efficacy of radial-isotropic transformations in robust regression and how our algorithms outperform baseline regression methods.

All experiments were done on a laptop computer with a 2.3 GHz Dual-Core Intel Core i5 CPU and 8 GB of RAM. We ran CVXPY's linear program solver for $\ell_1$-regression for linear functions.

**Recovering Linear Functions** We first show how our algorithm based on radial-isotropic position (Algorithm 1) compares to naive $\ell_1$ regression in exact recovery using an LP solver. As another baseline, we also ran $\ell_1$-regression with a normalization preprocessing step, where we normalize all points $(\mathbf{x}, y)$ to $\left(\frac{\mathbf{x}}{\|\mathbf{x}\|}, \frac{y}{\|x\|}\right)$. We did not run regression with an isotropic-transformation preprocessing step because this yields identical results as naive regression with no preprocessing.

We evaluated different transformations to the data on the following synthetic distribution. Define a mixture of Gaussians $\mathcal{D}_\mathbf{x} = \frac{1}{2}\mathcal{N}(\mathbf{e}_1, \frac{1}{d^2}I_d) + \frac{1}{2d}\sum_{i=1}^d \mathcal{N}(d\mathbf{e}_i, \frac{1}{d^2}I_d)$, where $\mathbf{e}_i$ denotes the $i$-th standard basis vector and $d = 30$. Let $\mathbf{w}^* = 9\mathbf{e}_2 + \sum_{i=1}^d \mathbf{e}_i$. For various noise levels $\eta$, consider the following $\eta$-Massart adversary: the labels for all $\mathbf{x}$ for which any coordinate is greater than $\frac{d}{2}$ are flipped to $-\mathbf{w}^* \cdot \mathbf{x}$ with probability $\eta$, and the labels for all other points are not flipped. Essentially, only the points *not* from $\mathcal{N}(\mathbf{e}_1, \frac{1}{d^2}I_d)$ are affected by Massart noise.

We measured exact parameter recovery rate, which captures how often the algorithm solves for $\mathbf{w}^*$ exactly. We varied the noise rate $\eta$ while running the methods with 120 samples from $\mathcal{D}_\mathbf{x}$. We also varied the sample size while keeping the noise $\eta = 0.25$. We ran 200 trials for each measurement of exact recovery rate and the error bars represent two standard deviations from the mean.

**Recovering ReLUs** For ReLUs, we used the same distribution as the experiments for linear functions to generate samples. We ran and compared constant-step-sized gradient descent on the empirical $\ell_1$-loss $\frac{1}{m}\sum_{i=1}^m |y_i - \mathrm{ReLU}(\mathbf{w}^* \cdot \mathbf{x}_i)|$ with different transformations to the data. We ran gradient descent since our separation oracle for the Ellipsoid method bears similarities with gradient descent. As seen in Lemma 3.1, this is due to the fact that our separating hyperplane is based on the gradient of the empirical $\ell_1$-loss of a subset of points.

The experiment is set up with $\eta = 0.4$, $\mathbf{w}^* = 9\mathbf{e}_2 + \sum_{i=1}^d \mathbf{e}_i$, $\mathbf{w}_0 = 0$, 240 samples from $\mathcal{D}_\mathbf{x}$, and gradient descent step size of one. For 'Original', we use a step size of $1/465$ to keep the magnitude of the points $\mathbf{x}_i$ comparable to that of the transformed points $\tilde{\mathbf{x}}_i$.

In Figure 2(a), 'Original' corresponds to naive gradient descent, while 'Normalized' has a normalization preprocessing step. The transformations of 'Isotropic' and 'Radial-isotropic' follow our algorithm for ReLUs from Section 3, where the transformation is only applied to the positive-side points of $\mathbf{w} \cdot \mathbf{x}_i \geq 0$ for the current hypothesis $\mathbf{w}$. The gradient is then calculated with the transformed points $\tilde{\mathbf{x}}_i$ and appropriately transformed back to the original space in order to update $\mathbf{w}$. The gradient descent updates under transformation $\mathbf{A}$ and step size $\alpha$ is the following:

$$\mathbf{w}' \leftarrow \mathbf{A}^{-1}\mathbf{w} \quad \text{then} \quad \mathbf{w} \leftarrow \mathbf{w} - \alpha \cdot (\mathbf{A}\nabla_{\mathbf{w}'}L'),$$

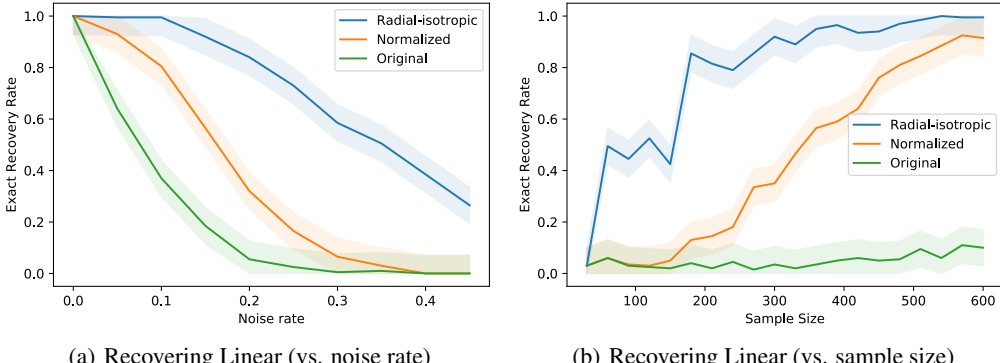

(a) Recovering Linear (vs. noise rate)    (b) Recovering Linear (vs. sample size)

Figure 1: Experiments for exact parameter recovery of linear functions on synthetic data. Exact recovery rate (y-axis) measures how often the algorithm outputs the true parameter out of 200 trials. We compare Algorithm 1 with naive $\ell_1$-minimization and $\ell_1$-minimization with normalized data. Error bars cover two standard deviations from the mean.

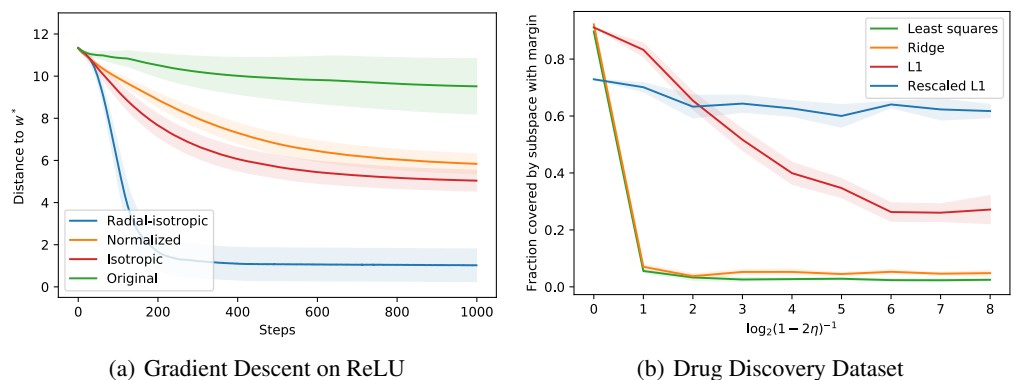

(a) Gradient Descent on ReLU    (b) Drug Discovery Dataset

Figure 2: On the left, we compare the performance of different transformations with gradient descent on synthetic data where we measure the $\ell_2$-distance to the optimal solution. On the right, we compare different regression methods applied to real data where the labels are artificially corrupted with $\eta$-Massart noise. Rescaled L1 represents Algorithm 1. We report the fraction of points that lie in the subspace generated by its output with margin.

where $L'$ denotes the empirical $\ell_1$-loss for the transformed subset of points. This update method is directly adapted from our Ellipsoid method.

**Drug Discovery Dataset**    The drug discovery dataset was originally curated by [40] and we used the same dataset as the one used in [14]. The dataset has a training and test set of 3084 and 1000 points of 410 dimensions. The $\eta$-Massart noise adversary corrupts the training data $(\mathbf{x}_i, y_i)$ so that all points are corrupted to flip labels to $-100y_i$ with probability $\eta$.

We compared $\ell_1$-regression with radial-isotropic transformation ('Rescaled L1') to other baseline methods, such as least squares and naive $\ell_1$-regression. For ridge regression, we optimized the regularization coefficient based on the uncorrupted data. We measure performance by computing the fraction of the test set that lies within the subspace generated by the output vector with a margin of 2.

**Results**    In Figure 1, our algorithm with radial-isotropic transformation outperforms other baseline methods in robustness with respect to the noise level and in efficiency with respect to the sample size. This is in line with the results of Theorem 1.2. In fact, our experiments on ReLUs also empirically demonstrate that radial isotropy significantly improves ReLU regression via gradient descent by making the dataset more robust to noise at each iteration. For the drug discovery dataset, although

$\ell_1$-regression with radial isotropy performs slightly worse than naive $\ell_1$-regression when there is minimal noise, it significantly outperforms the baseline methods at regimes of high noise levels.

## 5  Conclusion

In this work, we propose a generalization of the Massart (or bounded) noise model, previously studied in binary classification, to the real-valued setting. The Massart model is a realistic semi-random noise model that is stronger than uniform random noise or oblivious noise, but weaker than adversarial label noise. Our main result is an efficient algorithm for ReLU regression (and, in the process, also linear regression) in this model under minimal distributional assumptions. At the technical level, we provide structural conditions for $\ell_0$-minimization to be efficiently computable. A key conceptual idea enabling our efficient algorithms is that of transforming the dataset using radial-isotropic transformations. We empirically validated the effectiveness of radial-isotropic transformations for robustness via experiments on both synthetic and real data. In contrast to previous works on robust regression that require strong distributional assumptions, our framework and results may be seen as an intricate balance between slightly weakening the noise model yet affording generality in the underlying distribution.

## Acknowledgments and Disclosure of Funding

Ilias Diakonikolas is supported by NSF Medium Award CCF-2107079, NSF Award CCF-1652862 (CAREER), a Sloan Research Fellowship, and a DARPA Learning with Less Labels (LwLL) grant.

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
