## Supplementary Material

## A Omitted Proofs from Section 2

In the following, we provide all omitted proofs from Section 2.

### A.1 Proof of Lemma 2.1

We apply Theorem 1.5 and Proposition 2.7 of [1] to derive Lemma 2.1. [1] uses a generalized notion of radial isotropy where vectors $\{\mathbf{x}_1, \ldots, \mathbf{x}_n\} \subset \mathcal{S}^{d-1}$ lie in radial $c$-isotropic position if $\sum_{i=1}^n c_i \mathbf{x}_i \mathbf{x}_i^\top = I_d$ for $c \in \mathbb{R}^n$ such that $\|c\|_1 = d$. Here we are only interested in the case where $c = \frac{d}{n}\mathbb{1}$ which represents radial isotropy as defined in Definition 1.4.

The algorithm of Theorem 1.5, by definition, outputs a positive definite and symmetric matrix $\mathbf{A} := \left(\sum_{i=1}^d e^{t_i} \mathbf{x}_i \mathbf{x}_i^T\right)^{-1/2}$ where $t \in \mathbb{R}^d$, see, e.g., Section 2 of [1]. Thus, we focus on whether this transformation indeed yields a $\gamma$-approximate radial-isotropic transformation in polynomial time. For their algorithm to find a linear transformation $\mathbf{A}$ that puts the vectors in $\gamma$-approximate radial-isotropic position, we need to set $\varepsilon$ from Theorem 1.5 to be sufficiently small. We set $\sqrt{\varepsilon} = \frac{d}{n}\gamma$ so that $\|c_{\text{apx}} - \frac{d}{n}\mathbb{1}\|_2 \leq \sqrt{\varepsilon}$. For $\mathbf{v} \in \mathcal{S}^{d-1}$, their algorithm transforms the set of vectors such that we have the following relationship:

$$\sum_{i=1}^n \frac{d}{n}\langle \mathbf{x}_i, \mathbf{v}\rangle^2 \geq \sum_{i=1}^n \frac{d/n}{d/n + \sqrt{\varepsilon}}c_{\text{apx},i}\langle \mathbf{x}_i, \mathbf{v}\rangle^2$$
$$= \frac{d/n}{d/n + \sqrt{\varepsilon}} = \frac{1}{1 + \gamma} \geq 1 - \gamma.$$

Therefore, their algorithm yields a $\gamma$-approximate radial-isotropic transformation. It remains to show that $\|t^*\|_\infty$ is at most $\text{poly}(n, d, b)$: if so, Theorem 1.5 shows that we can efficiently compute an invertible linear transformation that puts general position points in $\gamma$-approximate radial-isotropic position in $\text{poly}(n, d, b, 1/\gamma)$-time.

We have the following bound on $\|t^*\|_\infty$ from Lemma 4.6 and Lemma 4.7 of [1].

$$\|t^*\|_\infty \leq \log \frac{n}{d} + (d-1)\log\left(\frac{32nd^2}{\Delta_S^{\min}}\right)$$

where $\Delta_S = \det((\mathbf{x}_i)_{i \in S})^2$ is the square determinant of a $d$-tuple of unit vectors ($|S| = d$) and $\Delta_S^{\min}$ is the smallest positive value of $\Delta_S$. Any positive determinant of $d$-tuple of vectors supported on $b$-bits must be at least 1, assuming each coordinate must be represented by an integer from 0 to $2^b - 1$. Then, after normalizing vectors so that we only consider unit vectors on $\mathcal{S}^{d-1}$, we have that $\Delta_S^{\min} \geq \frac{1}{(\sqrt{d}2^b)^d}$. Thus, $\|t^*\|_\infty = O(d\log n + d^2\log d + d^2 b)$ so we can get $1/2$-approximate radial-isotropic position in $\text{poly}(n, d, b)$-time.

By Lemma 4.3 of [1], the ratio between the largest and smallest eigenvalue of $\mathbf{A}$ is at most $\left(\frac{8n}{\delta^2}\right)^{(d-1)/2}$ where $\delta = \sqrt{\Delta_S^{\min}}/2d$. Thus the logarithm of the condition number of $\mathbf{A}$ is $O(d\log n + d^2\log d + d^2 b)$ so it is $\text{poly}(n, d, b)$. This concludes the proof of Lemma 2.1 for finding a $\gamma$-approximate radial-isotropic transformation in polynomial time for general position points.

## A.2 Proof of Lemma 2.2

Let $\mathbf{w}^*$ be the parameter corresponding to the $\ell_0$-minimizer. Denote the $\ell_1$-loss $\hat{L}(\mathbf{w}) = \frac{1}{m}\sum_{i=1}^{m}|y_i - f(\mathbf{w}\cdot\mathbf{x}_i)|$. Given the strict inequality in (⋆), for non-zero $r \in \mathbb{R}^d$, we have that

$$
\begin{aligned}
&m\big(\hat{L}(\mathbf{w}^* + \mathbf{r}) - \hat{L}(\mathbf{w}^*)\big) \\
&= \sum_{i=1}^{m}|f((\mathbf{w}^* + \mathbf{r})\cdot\mathbf{x}_i) - y_i| - \sum_{i=1}^{m}|f(\mathbf{w}^*\cdot\mathbf{x}_i) - y_i| \\
&= \sum_{y_i = f(\mathbf{w}^*\cdot\mathbf{x}_i)}|f((\mathbf{w}^* + \mathbf{r})\cdot\mathbf{x}_i) - f(\mathbf{w}^*\cdot\mathbf{x}_i)| + \sum_{y_i \neq f(\mathbf{w}^*\cdot\mathbf{x}_i)}|f((\mathbf{w}^* + \mathbf{r})\cdot\mathbf{x}_i) - y_i| - \sum_{y_i \neq h_{\mathbf{w}^*}(\mathbf{x}_i)}|f(\mathbf{w}^*\cdot\mathbf{x}_i) - y_i| \\
&\geq \sum_{y_i = f(\mathbf{w}^*\cdot\mathbf{x}_i)}|f((\mathbf{w}^* + \mathbf{r})\cdot\mathbf{x}_i) - f(\mathbf{w}^*\cdot\mathbf{x}_i)| - \sum_{y_i \neq f(\mathbf{w}^*\cdot\mathbf{x}_i)}|f((\mathbf{w}^* + \mathbf{r})\cdot\mathbf{x}_i) - f(\mathbf{w}^*\cdot\mathbf{x}_i)| > 0.
\end{aligned}
$$

Therefore, the $\ell_0$-minimizer $\mathbf{w}^*$ is also the $\ell_1$-minimizer $\arg\min_{\mathbf{w}\in\mathbb{R}^d}\frac{1}{m}\sum_{i=1}^{m}|y_i - h(\mathbf{x}_i)|$.

## A.3 Proof of Theorem 1.2

Given Lemma 2.1 and 2.2, we now prove the main theorem for robust linear regression based on Algorithm 1.

*Proof of Theorem 1.2.* Without loss of generality, assume $\mathbf{x}_i$'s are unit vectors. The linear function can be written as follows.

$$y = \mathbf{w}^*\cdot\mathbf{x} = (\mathbf{A}^{-1}\mathbf{w}^*)\cdot(\mathbf{A}\mathbf{x})$$

where $A$ denotes the $\gamma$-approximate radial-isotropic transformation where $\gamma = 1/2$. This means that the solution to the LP in Algorithm 1 returns $\hat{\mathbf{w}} = \mathbf{A}^{-1}\mathbf{w}^*$ given Lemma 2.2 is satisfied. Therefore, we output $A\hat{w}$ as the true direction of the original dataset.

The rest of the proof proves that the structural condition holds. By radial isotropy, for $\mathbf{r}\in\mathbb{R}^d$

$$\frac{1}{m}\sum_{i=1}^{m}|\mathbf{r}\cdot\tilde{\mathbf{x}}_i| \geq \frac{\|\mathbf{r}\|_2}{m}\cdot\sum_{i=1}^{m}(\frac{\mathbf{r}}{\|\mathbf{r}\|_2}\cdot\tilde{\mathbf{x}}_i)^2 \geq \frac{(1-\gamma)\|\mathbf{r}\|_2}{d}$$

Define $\tilde{\mathcal{D}}_{\mathbf{x}}$ on $\mathcal{S}^{d-1}$ to be the distribution $\mathcal{D}_{\mathbf{x}}$ after the transformation $\mathbf{x}\mapsto\frac{\mathbf{A}\mathbf{x}}{\|\mathbf{A}\mathbf{x}\|_2}$. By the VC inequality with $\tilde{O}(\frac{d}{\epsilon^2})$ samples, with high probability,

$$\sup_{\mathbf{r}\in\mathbb{R}^d}\left|\Pr_{\tilde{\mathbf{x}}\sim\tilde{\mathcal{D}}_{\mathbf{x}}}[|\mathbf{r}\cdot\tilde{\mathbf{x}}| > t] - \frac{1}{m}\sum_{i=1}^{m}\mathbb{1}\{|\mathbf{r}\cdot\tilde{\mathbf{x}}_i| > t\}\right| \leq \epsilon$$

since the VC dimension of $\mathcal{F} = \{\mathbb{1}_{\{|\mathbf{r}\cdot\mathbf{x}|>t\}} : \mathbf{r}\in\mathbb{R}^d, t\in\mathbb{R}\}$ is $O(d)$. By integration, we get

$$
\begin{aligned}
\frac{1}{m}\sum_{i=1}^{m}|\mathbf{r}\cdot\tilde{\mathbf{x}}_i| &= \int_0^{\infty}\left(\frac{1}{m}\sum_{i=1}^{m}\mathbb{1}\{|\mathbf{r}\cdot\tilde{\mathbf{x}}_i| > t\}\right)dt \\
&= \int_0^{\max_i|\mathbf{r}\cdot\tilde{\mathbf{x}}_i|}\left(\frac{1}{m}\sum_{i=1}^{m}\mathbb{1}\{|\mathbf{r}\cdot\tilde{\mathbf{x}}_i| > t\}\right)dt \\
&\leq \mathbb{E}_{\tilde{\mathbf{x}}\sim\tilde{\mathcal{D}}_{\mathbf{x}}}[|\mathbf{r}\cdot\tilde{\mathbf{x}}|] + \epsilon\cdot\max_i|\mathbf{r}\cdot\mathbf{x}_i| \\
&\leq \mathbb{E}_{\tilde{\mathbf{x}}\sim\tilde{\mathcal{D}}_{\mathbf{x}}}[|\mathbf{r}\cdot\tilde{\mathbf{x}}|] + \frac{\epsilon d}{1-\gamma}\left(\frac{1}{m}\sum_{i=1}^{m}|\mathbf{r}\cdot\tilde{\mathbf{x}}_i|\right)
\end{aligned}
$$

since $\max_i|\mathbf{r}\cdot\tilde{\mathbf{x}}_i| \leq \|\mathbf{r}\|_2$. Then we have the inequality:

$$\|\mathbf{r}\|_2 \leq \frac{(1-\gamma)d}{1-\gamma-\epsilon d}\mathbb{E}_{\tilde{\mathbf{x}}\sim\tilde{\mathcal{D}}_{\mathbf{x}}}[|\mathbf{r}\cdot\tilde{\mathbf{x}}|].$$

Then we get a lower bound for the uncorrupted samples.

$$\frac{1}{m}\sum_{i=1}^{m}|\mathbf{r}\cdot\tilde{\mathbf{x}}_i|\mathbb{1}\{y_i=\mathbf{w}^*\cdot\mathbf{x}_i\} = \int_0^\infty\left(\frac{1}{m}\sum_{i=1}^{m}\mathbb{1}\{|\mathbf{r}\cdot\tilde{\mathbf{x}}_i|>t\wedge y_i=\mathbf{w}^*\cdot\mathbf{x}_i\}\right)dt$$

$$\geq \mathbb{E}_{\tilde{\mathbf{x}}\sim\tilde{\mathcal{D}}_\mathbf{x}}[|\mathbf{r}\cdot\tilde{\mathbf{x}}|\mathbb{1}\{y=\mathbf{w}^*\cdot\mathbf{x}\}] - \epsilon\max_{\tilde{x}\in\text{supp}(\mathcal{D}_\mathbf{x})}|\mathbf{r}\cdot\tilde{x}|$$

$$\geq (1-\eta)\mathbb{E}_{\tilde{\mathbf{x}}\sim\tilde{\mathcal{D}}_\mathbf{x}}[|\mathbf{r}\cdot\tilde{\mathbf{x}}|] - \frac{(1-\gamma)\epsilon d}{1-\gamma-\epsilon d}\mathbb{E}_{\tilde{\mathbf{x}}\sim\tilde{\mathcal{D}}_\mathbf{x}}[|\mathbf{r}\cdot\tilde{\mathbf{x}}|].$$

We similarly obtain an upper bound for the corrupted samples of $y_i \neq \mathbf{w}^*\cdot\mathbf{x}_i$, so by setting $\epsilon = O(\frac{1-2\eta}{d})$, with $m = \tilde{O}(\frac{d^3}{(1-2\eta)^2})$ samples, the structural condition of Lemma 2.2 is satisfied for any non-zero $\mathbf{r}\in\mathbb{R}^d$ with high probability. Thus, with Lemma 2.1, this proves Theorem 1.2. □

## B  Omitted Proofs from Section 3

In the following, we provide all omitted proofs from Section 3.

### B.1  Proof of Lemma 3.1

We establish that we can find a separating hyperplane that separates $\mathbf{w}$ sufficiently far from $\mathbf{w}^*$. This makes sure the Ellipsoid method shrinks in volume while always containing a small ball around $\mathbf{w}^*$ that is never cut by a separating hyperplane.

*Proof of Lemma 3.1.* Define the empirical loss $\hat{L}(\mathbf{w}) = \frac{1}{m}\sum_{i=1}^{m}|y_i - f(\mathbf{w}\cdot\mathbf{x}_i)|$.

$$(\mathbf{w}_0-\mathbf{w}^*)\cdot\nabla\hat{L}(\mathbf{w}_0) = \frac{1}{m}\sum_{i=1}^{m}[\text{sgn}(f(\mathbf{w}_0\cdot\mathbf{x}_i)-y_i)(\mathbf{w}_0-\mathbf{w}^*)\cdot\nabla_\mathbf{w}f(\mathbf{w}_0\cdot\mathbf{x}_i)]$$

$$= \frac{1}{m}\sum_{i=1}^{m}[\text{sgn}(f(\mathbf{w}_0\cdot\mathbf{x}_i)-y_i)(\mathbf{w}_0-\mathbf{w}^*)\cdot\mathbf{x}_i f'(\mathbf{w}_0\cdot\mathbf{x}_i)]$$

$$= \frac{1}{m}\sum_{i:y_i=f(\mathbf{w}^*\cdot\mathbf{x}_i)}[(\mathbf{w}_0-\mathbf{w}^*)\cdot\mathbf{x}_i\cdot\text{sgn}(\mathbf{w}_0\cdot\mathbf{x}_i-\mathbf{w}^*\cdot\mathbf{x}_i)f'(\mathbf{w}_0\cdot\mathbf{x}_i)]$$

$$+ \frac{1}{m}\sum_{i:y_i\neq f(\mathbf{w}^*\cdot\mathbf{x}_i)}[(\mathbf{w}_0-\mathbf{w}^*)\cdot\mathbf{x}_i\cdot\text{sgn}(f(\mathbf{w}_0\cdot\mathbf{x}_i)-y_i)f'(\mathbf{w}_0\cdot\mathbf{x}_i)]$$

$$\geq \frac{1}{m}\sum_{i:y_i=f(\mathbf{w}^*\cdot\mathbf{x}_i)}[|(\mathbf{w}_0-\mathbf{w}^*)\cdot\mathbf{x}_i|f'(\mathbf{w}_0\cdot\mathbf{x}_i)]$$

$$- \frac{1}{m}\sum_{i:y_i\neq f(\mathbf{w}^*\cdot\mathbf{x}_i)}[|(\mathbf{w}_0-\mathbf{w}^*)\cdot\mathbf{x}_i|f'(\mathbf{w}_0\cdot\mathbf{x}_i)] \geq \Delta$$

where the third equality follows from monotonicity of $f$ and the last inequality follows from the inequality condition (†) on the set of samples. □

## C  Full Algorithms and Extended Results

In this section, we provide linear and ReLU regression algorithms that can handle distributions that have non-zero measure on a subspace. Since a non-trivial fraction of samples may concentrate on a particular subspace, there may not exist a transformation that puts the points into radial-isotropic position. In fact, the following condition is necessary and sufficient for the existence of such a transformation.

**Lemma C.1** (Lemma 4.19 of (53)). *Given a set of points $S\subseteq\mathbb{R}^d$, the following conditions are equivalent:*

1. *For any $\gamma > 0$, there exists an invertible linear transformation $\mathbf{A}$ such that $\mathbf{A}$ puts $S$ in $\gamma$-approximate radial-isotropic position.*

2. *For every $1 \leq k \leq d$, every $k$-dimensional subspace contains at most $k/d$-fraction of $S$.*

Given the condition above, there does not exist a radial-isotropic transformation for all non-zero points if there exists a $k$-dimensional subspace $V$ that contains more than $k/d$-fraction of the non-zero points. In this case, we use the following algorithmic result from (52) that efficiently computes a radial-isotropic transformation for the points that lie on the subspace $V$.

**Lemma C.2** (Theorem 1.4 of (52)). *There exists an algorithm that, given a set $S$ of $n$ points in $\mathbb{Z}^d \setminus \{0\}$ of bit complexity at most $b$ and $\delta > 0$, runs in $\mathrm{poly}(n, d, b, \log(1/\delta))$ time, and returns a subspace $V$ of $\mathbb{R}^d$ containing at least a $\dim(V)/d$-fraction of the points in $S$ and a linear transformation $\mathbf{A} : V \to V$ such that $\frac{1}{|S \cap V|} \sum_{\mathbf{x} \in S \cap V} (\frac{\mathbf{A}\mathbf{x}}{\|\mathbf{A}\mathbf{x}\|_2})(\frac{\mathbf{A}\mathbf{x}}{\|\mathbf{A}\mathbf{x}\|_2})^T = (1/\dim(V))I_V + O(\delta)$, where the error is in spectral norm.*

This algorithmic result relaxes the assumptions on the underlying distribution of Theorem 1.2 and 1.3 by allowing us to compute a radial-isotropic transformation for a set of points that may concentrate on a particular subspace. Hence we can make stronger claims. We describe the details of the algorithm and then state and prove our more generalized results (Theorem C.3 and C.4) below.

## C.1 Linear Regression

In general, we assume that $\mathcal{D}_{\mathbf{x}}$ is a distribution supported on $b$-bit integers such that $\Pr_{\mathbf{x} \sim \mathcal{D}_{\mathbf{x}}}[\mathbf{r} \cdot \mathbf{x} = 0] \leq 1 - \rho$, for all non-zero $\mathbf{r} \in \mathbb{R}^d$, where $\rho \in (0, 1]$ is a parameter.

Our ReLU learning algorithm leverages this algorithmic result. The main algorithmic idea is to apply radial-isotropic transformation iteratively on any concentrated subspace. For example, if there exists a subset of points lying in a $k$-dimensional subspace $V$, so that there does not exist a radial-isotropic transformation for the whole set of points, i.e., more than $k/d$-fraction of the points lie on $V$, then we can efficiently find such a subspace $V$ with a corresponding radial-isotropic transformation in its lower-dimensional space, using Lemma C.2. With this ingredient, we can compute $\mathrm{proj}_V \mathbf{w}^*$ using Algorithm 1 in $k$-dimensions. Similarly, we compute the orthogonal component of $\mathbf{w}^*$ on the orthogononal subspace $V^{\perp}$. Here it is important that we have enough points from the original set of samples that do not project to zero in $V^{\perp}$, since a significant portion lies on $V$.

The pseudocode of our learner is presented below, followed by a statement and proof of its properties. We denote by FORSTERTRANSFORM the algorithm that achieves Theorem 1.4 of (52).

---

**Algorithm 3** Linear function recovery via radial isotropy

1: **procedure** RECOVERLINEAR($(\mathbf{x}_i, y_i)_{i=1}^m \subset \mathbb{R}^d \times \mathbb{R}$)
2:     Run FORSTERTRANSFORM to find subspace $V$ and radial-isotropic transformation $\mathbf{A}$
3:     **if** $\dim(V) = d$ **then**
4:         $\tilde{S} \leftarrow \{(\frac{\mathbf{A}\mathbf{x}_i}{\|\mathbf{A}\mathbf{x}_i\|_2}, \frac{y_i}{\|\mathbf{A}\mathbf{x}_i\|_2}) : i \in [m] \text{ for } \mathbf{x}_i \neq 0\}$
5:         $\tilde{\mathbf{w}} \leftarrow \arg\min_{\mathbf{w} \in \mathbb{R}^d} \sum_{(\tilde{\mathbf{x}}, \tilde{y}) \in \tilde{S}} |\tilde{y} - \mathbf{w} \cdot \tilde{\mathbf{x}}|$ by solving the LP.
6:         **return** $\mathbf{A}\tilde{\mathbf{w}}$
7:     $S_V \leftarrow \{(\mathbf{x}_i, y_i) : i \in [m] \text{ where } \mathbf{x}_i \in V\}$
8:     Rotate $S_V$ into $\mathbb{R}^k$ and run Algorithm 1 with transformation $\mathbf{A}$.
9:     Let $\mathbf{w}$ be the output from the previous step, rotated back into $\mathbb{R}^d$.
10:     Let $V^{\perp}$ be the orthogonal subspace to $V$ in $\mathbb{R}^d$.
11:     $S_V^{\perp} \leftarrow \{(\mathrm{proj}_{V^{\perp}} \mathbf{x}_i, y_i - \mathbf{w} \cdot \mathrm{proj}_V \mathbf{x}_i) : i \in [m] \text{ where } \mathbf{x}_i \notin V\}$
12:     Rotate $S_V^{\perp}$ into $\mathbb{R}^{d-k}$ and run RECOVERLINEAR and rotate back to compute $\mathbf{w}^{\perp} \in \mathbb{R}^d$.
13:     **return** $\mathbf{w} + \mathbf{w}^{\perp}$
14: $m \leftarrow \tilde{O}(\frac{d^3}{\rho(1-2\eta)^2})$
15: Draw $m$ i.i.d. samples $(\mathbf{x}_i, y_i)_{i=1}^m$ with $\eta$-Massart noise
16: RECOVERLINEAR($(\mathbf{x}_i, y_i)_{i=1}^m$)

---

**Theorem C.3.** *Let $\mathcal{D}_{\mathbf{x}}$ be a distribution on $\mathbb{R}^d$ such that $\Pr_{\mathbf{x} \sim \mathcal{D}_{\mathbf{x}}}[\mathbf{r} \cdot \mathbf{x} = 0] \leq 1 - \rho$ for all non-zero $\mathbf{r} \in \mathbb{R}^d$. Let $\eta < 1/2$ be the upper bound on the Massart noise rate. Denote $\mathbf{w}^*$ the vector*

*representing the true linear function. There is an algorithm that draws $\tilde{O}(\frac{d^3}{\rho(1-2\eta)^2})$ samples, runs in $\mathrm{poly}(d, b, \rho^{-1}, (1-2\eta)^{-1})$ time, where $b$ is an upper bound on the bit complexity of the samples and parameters, and outputs $\mathbf{w}^*$ with probability at least $9/10$.*

*Proof.* Assume, for the sake of simplicity, that $\rho = 1$ so that the distribution does not concentrate on any lower-dimensional subspace. Then there always exists a radial-isotropic transformation $\mathbf{A}$ for any set of samples, as long as it has at least $d$ points, since all points are in general position. As we have shown in the proof of Theorem 1.2, when $\rho = 1$, the algorithm correctly returns $\mathbf{w}^*$ with high probability using $\tilde{O}(\frac{d^3}{(1-2\eta)^2})$ samples.

For $0 < \rho < 1$, the correctness of the algorithm follows from a standard divide-and-conquer argument, as long as each call to the algorithm is supplied with a sufficient number of (non-zero) samples. Thus, we only need to analyze the sample complexity and ensure each recursive call into $k$-dimensions receives enough samples as an input.

For the first iteration of RECOVERLINEAR in $\mathbb{R}^d$, if there exists $\mathbf{A}$ that puts the remaining non-zero points into radial isotropy, we only need to sample $\tilde{O}(\frac{d^3}{\rho(1-2\eta)^2})$ points from $\mathcal{D}_{\mathbf{x}}$. The factor of $\rho^{-1}$ appears because in the worst case we have $\rho$-fraction of the marginal distribution $\mathcal{D}_{\mathbf{x}}$ concentrating on $0$, so that $\rho$-fraction of the samples cannot be put into radial-isotropic position. Thus, we need $m_d := \tilde{O}(\frac{d^3}{\rho(1-2\eta)^2})$ many samples for $d$ dimensions if $\mathbf{A}$ exists. Then, similarly to Theorem 1.2, if $\mathbf{A}$ exists, $m_d$ many samples are sufficient for RECOVERLINEAR in $\mathbb{R}^d$ to find $\mathbf{w}^*$ with probability at least $9/10$. We now need to prove that the algorithm works with $m_d$ samples with probability at least $9/10$ even when $\mathbf{A}$ does not exist.

In the case that $\mathbf{A}$ does not exist, by Lemma C.1, there must exist a $k$-dimensional subspace $V$ that contains more than $k/d$-fraction of the points. Here, we apply the algorithm on the subset $S_V := \{(\mathbf{x}_i, y_i) : i \in [m] \text{ where } \mathbf{x}_i \in V\}$ in $\mathbb{R}^k$. In this subproblem, the number of samples is $|S_V| \geq (k/d)m_d \geq m_k$, and thus is sufficient to accurately compute the projection of $\mathbf{w}^*$ on $V$.

What remains is ensuring that $S_V^\perp$ has enough non-zero samples, despite more than $k/d$-fraction of the points projecting to zero on the orthogonal subspace $V^\perp$. In other words, we want to upper bound the probability that $S_V$ simultaneously contains more than $k/d$-fraction of the points and more than $m_d - \rho m_{d-k}$ points, for $1 \leq k < d$. By the union bound, we can simplify the following expression.

$$\Pr\left[\left(|S_V| \geq km_d/d\right) \wedge \left(|S_V| \geq m_d - \rho m_{d-k}\right) \text{ for } 1 \leq k < d\right]$$
$$\leq \Pr[|S_V| \geq (1 - \rho + \rho k/d)m_d \text{ for } 1 \leq k < d]$$
$$\leq \sum_{k=1}^{d-1} \Pr[|S_V| \geq (1 - \rho + \rho k/d)m_d] \leq (d-1)\Pr[|S_V| \geq (1 - \rho + \rho/d)m_d].$$

If $\rho \geq 1/2$, Hoeffding's inequality bounds from above this quantity by $(d-1)\exp(-\frac{m_d}{2d^2})$. In the case that $1 - \rho \geq 1/2$, we have that $D_{\mathrm{KL}}(1 - \rho + \delta || 1 - \rho) \geq \frac{\delta^2}{2\rho(1-\rho)}$, so the Chernoff bound yields the following inequality:

$$\sum_{k=1}^{d-1} \Pr[|S_V| \geq (1 - \rho + \frac{\rho k}{d})m_d] \leq \sum_{k=1}^{d-1} \exp\left(-\frac{(\rho k/d)^2}{2\rho(1-\rho)}m_d\right)$$
$$\leq (d-1)\exp\left(-\frac{\rho}{2d^2}m_d\right).$$

Thus, with $m_d = \tilde{O}(\frac{d^3}{\rho(1-2\eta)^2})$, we can guarantee that any heavy subspace $V$ with more than $k/d$-fraction of the points will not contain too many samples, meaning that there will be $\tilde{O}(\frac{d^3}{(1-2\eta)^2})$ non-zero points in $S_V^\perp$ to compute a radial-isotropic transformation if one exists. Furthermore, the error probability we calculated above may accumulate over at most $d$ recursive calls. Since the error we have above is bounded in terms of $\exp(-\frac{d^3}{(1-2\eta)^2})$, after applying the union bound, we can still ensure that the algorithm finds $\mathbf{w}^*$ with high probability. $\square$

## C.2 ReLU Regression

In this subsection, we provide generalized results for ReLUs. The main idea remains similar to that of linear regression of Theorem C.3. If there exists a concentrated subspace $V$, we work in the lower-dimensional subspace to obtain a separation oracle between $\text{proj}_V \mathbf{w}^*$ and the query $\text{proj}_V \mathbf{w}_0$. In the case that $\text{proj}_V \mathbf{w}^* = \text{proj}_V \mathbf{w}_0$, we look at the orthogonal subspace $V^\perp$ to find a separating hyperplane there. We describe this procedure and theoretical guarantee in full detail below.

---

**Algorithm 4** Separation oracle sub-procedure

---

1: **Input:** $\{(\mathbf{x}_i, y_i)\}_{i=1}^m$ with Massart noise and query $\mathbf{w}_0$.
2: **Output:** If $\mathbf{w}_0 \in \mathcal{B}(\mathbf{w}^*, \Delta)$, return "Yes".
3:                 If not, return a separating hyperplane between $\mathbf{w}_0$ and $\mathcal{B}(\mathbf{w}^*, \Delta/2)$.
4: **procedure** SEP$((\mathbf{x}_i, y_i)_{i=1}^m, \mathbf{w}_0)$
5:      **if** $\text{ReLU}(\mathbf{w}_0 \cdot \mathbf{x})$ fits at least $\frac{m}{2}$ points **then**
6:          **return** "Yes"
7:      Define $S = \{(\mathbf{x}_i, y_i) : \mathbf{w}_0 \cdot \mathbf{x}_i \geq 0, \mathbf{x}_i \neq 0 \text{ for } i \in [m]\}$.
8:      Run FORSTERTRANSFORM on $S_\mathbf{x}$ to find subspace $V$ and radial-isotropic transformation $\mathbf{A}$
9:      **if** $\dim(V) = d$ **then**
10:          $\mathbf{r} = \frac{1}{|S|} \sum_{(\mathbf{x}_i, y_i) \in S} \frac{\mathbf{A}\mathbf{x}_i}{\|\mathbf{A}\mathbf{x}_i\|_2} \cdot \text{sgn}(\mathbf{w}_0 \cdot \mathbf{x}_i - y_i)$
11:          **return** separating hyperplane $\mathbf{A}^{-1}\mathbf{r} \cdot (\mathbf{w}_0 - \mathbf{w}) = 0$
12:      Rotate $\{(\mathbf{x}, y) \in S : \mathbf{x} \in V\}$ and the query $\text{proj}_V \mathbf{w}_0$ into $\mathbb{R}^k$ and run SEP on them
13:      **if** SEP returns a hyperplane in $\mathbb{R}^k$ **then**
14:          **return** the hyperplane rotated back and extended into $\mathbb{R}^d$ so that it is orthogonal to $V$
15:      Let $V^\perp$ be the orthogonal subspace to $V$ in $\mathbb{R}^d$.
16:      Define $S_V^\perp = \{(\text{proj}_{V^\perp} \mathbf{x}, \; y - \text{proj}_V \mathbf{w}_0 \cdot \text{proj}_V \mathbf{x}) : (\mathbf{x}, y) \in S \setminus V\}$.
17:      Rotate $S_V^\perp$ and the query $\text{proj}_{V^\perp} \mathbf{w}_0$ into $\mathbb{R}^{d-k}$ and run SEP on them
18:      **if** SEP returns a hyperplane in $\mathbb{R}^{d-k}$ **then**
19:          **return** the hyperplane rotated back and extended into $\mathbb{R}^d$ so that it is orthogonal to $V^\perp$
20:      **return** "Yes"

---

**Theorem C.4.** *Let $\mathcal{D}_\mathbf{x}$ be a distribution on $\mathbb{R}^d$ such that $\Pr_{\mathbf{x} \sim \mathcal{D}_\mathbf{x}}[\mathbf{r} \cdot \mathbf{x} = 0 | \mathbf{w} \cdot \mathbf{x} \geq 0] \leq 1 - \rho$ and $\Pr_{\mathbf{x} \sim \mathcal{D}_\mathbf{x}}[\mathbf{w} \cdot \mathbf{x} \geq 0] \geq \lambda$ for all non-zero $\mathbf{r}, \mathbf{w} \in \mathbb{R}^d$. Let $\eta < 1/2$ be the upper bound on the Massart noise rate. Denote $\mathbf{w}^*$ the parameter vector of the target ReLU. There is an algorithm that draws $\tilde{O}\left(\frac{d^3}{\rho \lambda^2 (1-2\eta)^2}\right)$ samples, runs in time $\text{poly}(d, b, \rho^{-1}, \lambda^{-1}, (1-2\eta)^{-1})$, and outputs $\mathbf{w}^*$ with probability at least $9/10$.*

*Proof.* Let $S = \{(\mathbf{x}_i, y_i) : \mathbf{w}_0 \cdot \mathbf{x}_i \geq 0, \mathbf{x}_i \neq 0 \text{ for } i \in [m]\}$ and denote $S_\mathbf{x}$ to be the set of covariates $\mathbf{x}_i$'s of $S$. Given $\mathbf{w}_0 \neq \mathbf{w}^*$, we have proved that there exists a separating hyperplane for the case when a radial-isotropic transformation $\mathbf{A}$ exists for $S_\mathbf{x}$ in the proof of Theorem I.3. If such $\mathbf{A}$ does not exist, this necessarily means that there exists a $k$-dimensional subspace $V$ that contains at least $\frac{k}{d}$-fraction of $S_\mathbf{x}$.

Given $V$, the points on $V$ are not affected by the orthogonal component of $\mathbf{w}^*$, but only $\text{proj}_V \mathbf{w}^*$. This provides a basis for a divide-and-conquer approach, where we run the separation oracle on this smaller subspace of dimension $k < d$. So, with appropriate rotation and rescaling, we can represent the points of $S_\mathbf{x}$ on $V$ and all projections onto $V$ in $k$-dimensions using $b$-bits.

The base case of $d = 1$ has a trivial radial-isotropic transformation, which can be any non-zero scalar, so the previous theorem we proved applies. Using strong induction, we assume that the separation oracle returns a correct output for the points on $V$ and $\text{proj}_V \mathbf{w}_0$. If SEP returns "Yes", then it must be that $\text{proj}_V \mathbf{w}^* = \text{proj}_V \mathbf{w}_0$. To find a separating hyperplane, we can then find one with respect to the orthogonal subspace $V^\perp$. Since $y_i = \mathbf{w}^* \cdot \mathbf{x}_i = \text{proj}_V \mathbf{w}^* \cdot \text{proj}_V \mathbf{x}_i + \text{proj}_{V^\perp} \mathbf{w}^* \cdot \text{proj}_{V^\perp} \mathbf{x}_i$, we can reduce this $d$ dimensions into $d - k$ and run SEP in a smaller subspace. The recursive call returns a correct separating hyperplane for the projections by strong induction, because if it returns "Yes", then $\mathbf{w}^* = \mathbf{w}_0$; but this cannot happen by our first if statement that checks majority.

When our recursive call does return a separating hyperplane in $V$, that means that the $k$-dimensional hyperplane separates $\text{proj}_V \mathbf{w}_0$ and $\mathcal{B}(\text{proj}_V \mathbf{w}^*, \Delta/2)$. Then the $d$-dimensional hyperplane, which

contains $k$-dimensional hyperplane and is orthogonal to $V$, separates $\mathbf{w}_0$ and $\mathcal{B}(\mathbf{w}^*, \Delta/2)$. Similarly, the separating hyperplane in $V^\perp$ yields a $d$-dimensional hyperplane that separates $\mathbf{w}_0$ and $\mathcal{B}(\mathbf{w}^*, \Delta/2)$.

The sample complexity to guarantee a correct separation oracle at all iterations follows similarly to that of Theorem C.3, and the number of iterations of the ellipsoid method is bounded by $\mathrm{poly}(d, b, (1 - 2\eta)^{-1})$, as the condition number of the linear transformations is bounded by $\mathrm{poly}(d, b, (1 - 2\eta)^{-1})$ by Proposition 2.2 of (52). This completes the proof of Theorem C.4. $\qquad\square$

# D   PAC Learning Linear Functions

In this section, we provide an algorithm for PAC learning linear functions in the presence of Massart noise.

For linear functions, if $\mathcal{D}_\mathbf{x}$ lies within a subspace of $\mathbb{R}^d$, then $\mathbf{w}^*$ would not information-theoretically idenitifiable. Thus, it is required that $\Pr_{\mathbf{x} \sim \mathcal{D}_\mathbf{x}}[\mathbf{r} \cdot \mathbf{x} = 0] \leq 1 - \lambda < 1$ for our exact recovery results. However, even when these assumptions are violated and the problem is non-identifiable, we provide a PAC learning guarantee for the linear case. Specifically, Theorem D.1 allows us to avoid any assumptions on the underlying distribution and output a function arbitrarily close the true function, even when exact recovery is information-theoretically impossible.

**Theorem D.1** (PAC Learning Linear Functions). *Let $\mathcal{D}_\mathbf{x}$ be a distribution on $\mathbb{R}^d$ with bit complexity $b$ and let $\eta < 1/2$ be the upper bound on the Massart noise rate. Denote by $\mathbf{w}^*$ the true target vector. There is an algorithm that draws $\tilde{O}(\frac{d^4 b^3}{\epsilon^3 (1-2\eta)^2})$ samples, runs in $\mathrm{poly}(d, b, \epsilon^{-1}, (1-2\eta)^{-1})$ time, and outputs $\hat{\mathbf{w}}$ such that $\Pr_{\mathbf{x} \sim \mathcal{D}_\mathbf{x}}[\hat{\mathbf{w}} \cdot \mathbf{x} \neq \mathbf{w}^* \cdot \mathbf{x}] \leq \epsilon$ with probability at least $9/10$.*

The PAC learning algorithm is similar to Algorithm 1, except instead of a radial-isotropic transformation, we run a spectral outlier removal procedure on the $m$ samples and solve the LP with the remaining inlier points only. This procedure, similarly to radial-isotropic transformations, minimizes the influence of points that are abnormally far from other points and thus nullifies the adversarial noise added to such points. We use the following definition of an outlier.

**Definition D.2** (Outlier). *We call a point $\mathbf{x}$ in the support of the distribution $\mathcal{D}_\mathbf{x}$ a $\beta$-outlier, if there exists a vector $\mathbf{w} \in \mathbb{R}^d$ such that $\langle \mathbf{w}, \mathbf{x} \rangle^2 > \beta \mathbb{E}_{\mathbf{x} \sim \mathcal{D}_\mathbf{x}}[\langle \mathbf{w}, \mathbf{x} \rangle^2]$.*

Theorem D.1 makes use of the following spectral outlier removal procedure by (24).

**Lemma D.3** (Theorem 3 of (24)). *Using $\tilde{O}(\frac{d^2 b}{\alpha})$ samples from $\mathcal{D}_\mathbf{x}$ where $\alpha > 0$, one can identify with high probability an ellipsoid $E$ such that $\Pr_{\mathbf{x} \sim \mathcal{D}_\mathbf{x}}[\mathbf{x} \in E] \geq 1 - \alpha$ and $\mathcal{D}_\mathbf{x}|_E$ has no $\tilde{O}(\frac{db}{\alpha})$-outliers.*

Lemma D.3 shows that there is an efficient algorithm that can preprocess any distribution supported on $b$-bit integers so that no large outliers exist. With this subroutine, we can achieve the same result of radial-isotropic transformation in Algorithm 1 with an arbitrary distribution, albeit with a sample complexity dependent on the bit complexity. So instead of radial isotropy, we run the outlier removal procedure with $\alpha \leftarrow \epsilon/2$.

*Proof of Theorem D.1.* Let $\mathcal{D}_\mathbf{x}$ be a distribution on $\mathbb{R}^d$ such that $\Pr_{\mathbf{x} \sim \mathcal{D}_\mathbf{x}}[\mathbf{r} \cdot \mathbf{x} = 0] \leq 1 - \lambda$ for all non-zero vector $\mathbf{r} \in \mathbb{R}^d$. Although $\lambda$ is not a quantity we know in the PAC learning setting, we will act as if we know what $\lambda$ is, as we will later replace it with $\epsilon$.

First, we prove that there is a $\mathrm{poly}(d, b, \lambda^{-1}, (1-2\eta)^{-1})$-time algorithm that draws $\tilde{O}(\frac{d^4 b^3}{\lambda^3 (1-2\eta)^2})$ samples and learns linear functions exactly with high probability.

By applying the outlier removal procedure with $\alpha = \lambda/2$ from Lemma D.3, with high probability, the new ellipsoid-truncated distribution $\mathcal{D}_\mathbf{x}^E$ has no $\tilde{O}(\frac{db}{\lambda})$-outliers. Since the outputted ellipsoid $E$ has mass at least $1 - \lambda/2$, $\mathcal{D}_\mathbf{x}'|_E$ remains fully $d$-dimensional.

We then use the VC inequality as in the proof for Theorem 1.2. Assume the $m$ samples here are the number of samples remaining after outlier removal.

$$\frac{1}{m}\sum_{i=1}^{m}|\mathbf{r}\cdot\mathbf{x}_i|\mathbb{1}\{y_i=\mathbf{w}^*\cdot\mathbf{x}_i\} = \int_0^\infty\left(\frac{1}{m}\sum_{i=1}^{m}\mathbb{1}\{|\mathbf{r}\cdot\mathbf{x}|>t\wedge y=\mathbf{w}^*\cdot\mathbf{x}\}\right)dt$$
$$\geq \mathbb{E}_{\mathcal{D}_{\mathbf{x}}^E}[|\mathbf{r}\cdot\mathbf{x}|\mathbb{1}\{y=\mathbf{w}^*\cdot\mathbf{x}\}] - \epsilon\max_{\mathbf{x}\in E}|\mathbf{r}\cdot\mathbf{x}|$$
$$\geq (1-\eta-\epsilon(2\beta)^{1.5})\mathbb{E}_{\mathcal{D}_{\mathbf{x}}^E}[|\mathbf{r}\cdot\mathbf{x}|]$$

The last inequality comes from following lemma. Let $x$ be a point in the support of a one-dimensional distribution $\mathcal{D}$ and let $X$ be the random variable defined by $\mathcal{D}$. If $x^2 \leq \beta\mathbb{E}[X^2]$, then $|x| \leq (2\beta)^{1.5}\mathbb{E}[|X|]$. This is because, wlog, we can assume $x \leq 1$ by normalizing since $X$ is bounded above. Then $\mathbb{E}[X^2] > 1/\beta$ so we have that $\Pr[X^2 > \frac{1}{2\beta}] > \frac{1}{2\beta}$. In other words, $\Pr[|X| > \frac{1}{\sqrt{2\beta}}] > \frac{1}{2\beta}$, so $\mathbb{E}[|X|] > (\frac{1}{2\beta})^{1.5}$. Therefore, $|x| \leq (2\beta)^{1.5}\mathbb{E}[|X|]$.

Ultimately, we want the RHS $1 - \eta - \epsilon(2\beta)^{1.5}$ to be greater than $\frac{1}{2}$. Similarly, we can guarantee $\frac{1}{m}\sum_{i=1}^{m}|\mathbf{r}\cdot\mathbf{x}_i|\mathbb{1}\{y_i\neq\mathbf{w}^*\cdot\mathbf{x}_i\}$ to be less than $\frac{1}{2}\mathbb{E}_{\mathcal{D}_{\mathbf{x}}^E}[|\mathbf{r}\cdot\mathbf{x}|]$. For this to hold, we need $\epsilon < \frac{1-2\eta}{2(2\beta)^{1.5}}$. Therefore, $1/\epsilon^2 = O(\frac{\beta^3}{(1-2\eta)^2})$ and $\beta = \tilde{O}(\frac{db}{\lambda})$, so we need at least $m = \tilde{O}(\frac{d^4b^3}{\lambda^3(1-2\eta)^2})$ samples for exact recovery.

Finally, we can replace the anti-concentration parameter $\lambda$ with $\epsilon$. This concludes the proof for PAC learning. $\square$

## E   Oblivious Noise and Massart Noise

In this section, we provide a formal comparison between the oblivious noise model and the Massart noise model for regression. We first define oblivious noise as defined in previous works.

**Definition E.1** (Oblivious Noise). *Given $0 \leq \eta < 1$, the oblivious adversary operates as follows. The algorithm specifies $m$ and the adversary corrupts the clean labels by adding sparse additive noise $\mathbf{b} = [b_1, b_2, \ldots, b_m]^T$ with no knowledge of the covariates $\mathbf{x}_i$ such that*

$$y_i = \mathbf{w}^*\cdot\mathbf{x}_i + b_i$$

*where $\mathbf{x}_i \sim \mathcal{D}_{\mathbf{x}}$, $\|\mathbf{b}\|_0 \leq \eta m$, and $\mathbf{b}$ is independent of $\mathbf{x}_i$'s and $\mathbf{w}^*$.*

Note that the breakdown point of a Massart adversary is $1/2$, yet the breakdown point of an oblivious adversary is not necessarily so. For instance, (47; 22) recover $\mathbf{w}^*$ even when the noise rate $\eta$ is arbitrarily close to 1.

We now establish the relationship between this model and the Massart noise model for regression.

**Lemma E.2.** *Given $m$ clean samples $(\mathbf{x}_i, f(\mathbf{x}_i))_{i=1}^{n}$ to corrupt, an Massart adversary of noise rate $\eta + \sqrt{\frac{\log(1/\delta)}{2m}}$ can simulate an oblivious adversary of noise rate $\eta$ with probability at least $1-\delta$.*

*Proof.* Since $\mathbf{x}_i$ is sampled i.i.d. from $\mathcal{D}_{\mathbf{x}}$ and the oblivious adversary chooses $\mathbf{b}$ without any knowledge of $\mathbf{x}_i$—hence the independence—adding the corruption vector $\mathbf{b}$ to the labels is equivalent to adding the corruption vector $\mathbf{U}\mathbf{b}$ where $\mathbf{U} \in \mathbb{R}^{m\times m}$ is a permutation matrix chosen uniformly at random and independently of $\mathbf{x}_i$ and $\mathbf{w}^*$. Therefore, for each fixed sample $(\mathbf{x}_i, f(\mathbf{x}_i))$, the label is corrupted by a random non-zero entry of $\mathbf{b}$ with probability at most $\eta$.

Given the above alternative description of oblivious noise, we can directly compare Massart noise with oblivious noise. Intuitively it is not difficult to see that an $\eta$-Massart adversary can simulate an $\eta$-oblivious adversary in expectation. After inspecting which of the $m$ samples can be corrupted after randomness, on average, there will be $\eta m$ labels that can be corrupted and the Massart adversary can add non-zero entries of $\mathbf{b}$ uniformly at random to these labels. This simulates the $\eta$-oblivious adversary as long as the Massart adversary can corrupt at least $\eta m$ samples which is determined probabilistically.

Because the oblivious adversary has the ability to deterministically choose how many labels to corrupt, an $\eta$-Massart adversary would not be able to simulate an $\eta$-oblivious adversary with high probability.

However, this is easily bounded by Hoeffding's inequality such that an $\left(\eta + \sqrt{\frac{\log(1/\delta)}{2m}}\right)$-Massart adversary can corrupt at least $\eta m$ samples with probability at least $1 - \delta$ and therefore can simulate an oblivious adversary of noise rate $\eta$. This means that, with more samples, an $(\eta + o(1))$-Massart adversary is stronger than an $\eta$-oblivious adversary with high probability $\qquad\square$

## F    Comparison to Chen et al.

First, we note that our work focuses on robust ReLU regression while the concurrent work of (10) focuses on robust linear regression and contextual bandits in the online setting. Yet even with different goals and directions, there is noticeable overlap between our results in robust linear regression from Section 2 and their results of robust linear regression in the offline setting from Section 5 and 6 of (10). In an online fashion, the adversary of (10) is allowed to corrupt the label $y_i$ arbitrarily with probability $\eta$ based on $\mathbf{x}_i$ and the previous samples $(\mathbf{x}_1, y_1), \ldots, (\mathbf{x}_{i-1}, y_{i-1})$. Furthermore, the covariates $\mathbf{x}_i$ do not necessarily have to come from a distribution and may be chosen adversarially at each round and hence the "distribution-free" robustness. Without random observation noise in the clean labels, the setting is similar to the Massart noise model in our work. In fact, the offline version of their adversary is identical to the Massart adversary, except the assumption on the covariates $\mathbf{x}_i$. We compare their algorithmic results and analysis for the realizable (offline) setting considered in this work below.

We first state their offline regression result adapted to the realizable setting considered in this paper. Refer to Theorem 6.11 from Section 6 of (10) for the following result achieved through using one of the two approaches, depending on the value of $\eta$.

**Theorem F.1** (Theorem 6.11 of (10) for the realizable setting)**.** *Suppose* $\|\mathbf{x}_i\|_2 \leq 1$, $\|\mathbf{w}^*\|_2 \leq R$ *for all rounds* $i \in [n]$ *and* $n = \Omega(\log(\min(n, d)/\delta))$. *Define* $\rho^2 = \frac{1-2\eta}{2\eta}$, $\Sigma_n = (1/n)\sum_{i=1}^n \mathbf{x}_i \mathbf{x}_i^T$, *and* $\|\mathbf{x}\|_\Sigma := \langle \mathbf{x}, \Sigma\mathbf{x} \rangle$. *There is a* $\text{poly}(n, d)$ *time algorithm which takes as input* $(\mathbf{x}_1, y_1), \ldots, (\mathbf{x}_n, y_n)$ *where the labels are only corrupted by* $\eta$-*Massart noise and outputs a vector* $\hat{\mathbf{w}}$ *which achieves*

$$\|\hat{\mathbf{w}} - \mathbf{w}^*\|_{\Sigma_n} \leq O\left(\frac{R}{\min(1, \rho^2)}\sqrt[4]{\frac{\eta \log(\min(n, d)/\delta)}{n}}\right)$$

*with probability at least* $1 - \delta$.

For the realizable setting where there is no observation noise, the result above yields a significantly weaker guarantee. Efficient exact recovery must output a vector $\hat{\mathbf{w}}$ such that $\|\hat{\mathbf{w}} - \mathbf{w}^*\| \leq \epsilon$ in time polynomial in $\log(1/\epsilon)$, not $1/\epsilon$. However, they do not achieve efficient exact recovery since it takes $\text{poly}(1/\epsilon)$ many samples to achieve error $\epsilon$. Futhermore, its guarantee depends on concentration properties of the covariates as denoted by $\Sigma_n$. Another major difference is that their algorithm incurs a polynomial dependence on $R$ which is unnecessary under our problem setting.

## Supplementary Material References

[52]  DIAKONIKOLAS, I., KANE, D. M., AND TZAMOS, C. Forster decomposition and learning halfspaces with noise. *CoRR abs/2012.09720* (2021).

[53]  HOPKINS, M., KANE, D. M., LOVETT, S., AND MAHAJAN, G. Point location and active learning: Learning halfspaces almost optimally. In *61st IEEE Annual Symposium on Foundations of Computer Science, FOCS 2020* (2020), pp. 1034–1044.