# OpenReview forum: "ReLU Regression with Massart Noise"
_NeurIPS.cc/2021/Conference — NeurIPS 2021 Poster_

### Official Review · Reviewer_8NsY · 2021-07-03

**Rating:** 6
**Confidence:** 4

**Summary:**

This paper studies regression problems in presence of Massart (bounded) noise, where for each sample x_i, the adversary is allowed to change its label/response f(x_i) to an arbitrary value with probability less or equal to eta. Specifically, the authors focus on two problems: linear regression and ReLU regression. For both cases, they make a ``mild yet necessary assumption on the data, that the distribution is not concentrated entirely on any linear subspace; and provide algorithms which exactly recover the true function in polynomial time with constant probability.

**Limitations And Societal Impact:**

yes

**Main Review:**

I feel the paper is well-written but I have a few concerns on the theoretical guarantee.

- In Theorem 1.2, it turns out that the sample complexity for robust linear regression is suboptimal. In particular, even when the noise rate eta = 0, it is given by O(d^3) which is worse than that of noiseless regression. It also seems unpleasant to have bit complexity dependence in running time.

- In Theorem 1.3, I cannot follow the assumption that $P( w \cdot x \geq 0 ) \geq \lambda$ for all w.

- What is the computational cost to obtain a radial-isotropic transformation matrix A? This turns out to be a crucial algorithmic component but I did not find a concrete discussion. Note that since you require unit data norm and identity covariance simultaneously, finding such transformation is nontrivial.

- The distributional assumption is informally stated throughout. Can you provide a fews examples that satisify the condition, say Gaussian distributions or uniform distributions?

- I may miss some important messages, but if the data distribution were uniform, then would it be necessary to perform radial-isotropic transformation? My understanding is not being necessary, in which case the technical novelty appears vacuous.

**Time Spent Reviewing:**

5

---

> ### Author Response · Authors · 2021-08-10
> **Author response for Reviewer 8NsY**
>
> We thank the reviewer for the helpful suggestions and comments. Below we respond to the main concerns/questions from the reviewer.
>
> **Sample Complexity:** As the reviewer has mentioned, the sample complexity of our algorithm for linear regression with Massart noise is quantitatively higher, compared to the noiseless case (corresponding to $\eta=0$). While it is an interesting question to improve on the sample complexity of our algorithm, we emphasize that our work gives the *first polynomial sample and time* algorithm with strong recovery guarantees for ReLUs under minimal distributional assumptions.
>
> **Bit complexity Dependence in Runtime:** As the reviewer has noted, the runtime of our algorithm depends on the bit complexity $b$ of the points. This is a common theme in *any* algorithm that uses linear programming as a subroutine. For example, even for the basic problem of learning a linear separator *without noise*, all known algorithms require such a bit complexity dependence. In fact, such a dependence cannot be removed without a strongly polynomial time algorithm for arbitrary linear programs -- a major open problem in computer science. In our specific setting, we solve a linear program (or use the ellipsoid method) to perform exact recovery, which incurs a runtime polynomial in $\log(1/\epsilon)$, not $1/\epsilon$. Given our learning task, we cannot remove the dependence on $b$ without a strongly polynomial algorithm for LPs. However, it is important to note that the sample complexity of our algorithms does not depend on the bit complexity, as a consequence of using radial isotropy. In fact, spectral outlier removal procedures incur a b dependence in the sample complexity, as demonstrated in our PAC learning results (Appendix D). Therefore, we highlight that the bit complexity dependence on the runtime is not unusual and rather that the lack of $b$ in the sample complexity is an achievement.
>
> **Distributional Assumptions and Examples:** The distributional assumption we require, i.e., that $\Pr[w \cdot x \geq 0] \geq \lambda$, means that there exists at least some non-trivial probability mass on any homogeneous halfspace. For example, the parameter $\lambda$ would be $1/2$ for any distribution symmetric around the origin. This is an extremely mild condition that is satisfied by a wide range of distributions. For example, it is satisfied by any mean-zero distribution with non-degenerate covariance matrix. In contrast, virtually all prior results on linear or ReLU regression require at least some non-trivial concentration (tail bounds), anti-concentration, and anti-anti-concentration properties.
>
> **Technical Novelty of the use of Radial-isotropic Transformations:** The central question of our paper is whether there exists realistic label noise models in which efficient learning is possible *without strong distributional assumptions*. We know that stronger adversarial noise models, such as the strong contamination model, have computational hardness results as described in the introduction (Lines 27-34). Moreover, for the Massart noise model, there has been recent work (see references 9, 12 in our paper) that learn halfspaces in a distribution-independent fashion. Analogously, we provide ReLU regression results with only mild assumptions that do not require any strong tail bounds or concentration bounds.
>
> Indeed, as the reviewer points out, if the data is uniformly distributed on the unit sphere, then we do not need to perform radial-isotropic transformations since the data is already well-behaved. However, this is not the case for most distributions, even after normalization. Thus, the use of radial-isotropic transformations allows us to generalize our regression method to apply beyond such well-behaved distributions.
>
> **Computational Cost of Radial-Isotropic Transformation:** Computing a radial-isotropic transformation $A$ can be done in polynomial time as specified in Lemma 2.1. Because we do not need an exact transformation but only an approximate one where $\gamma = 1/2$, we can use previously established algorithmic results in computing this approximate transform. We describe how to compute $A$ using the algorithmic results of Artstein-Avidan et al. and prove this lemma in Appendix A.

---

> > ### Comment · Reviewer_8NsY · 2021-08-11
> > **reviewer comments**
> >
> > Thank you for the response. It addressed all my concerns.

---

### Official Review · Reviewer_Dpt6 · 2021-07-15

**Rating:** 7
**Confidence:** 3

**Summary:**

The paper considers the problem of linear and ReLu regression in the Massart noise model, where an adversary is allowed to change the label to an arbitrarily value with some probability at most $\eta <1/2$. The work develops an efficient algorithm that achieves exact parameter recovery in this model under mild assumptions on the underlying distribution.

**Main Review:**

**Quality and Clarity**:

-  One of the strong points if the paper is that it is very well-written and well organized. The ideas are presented clearly with intuitive explanations, and the proofs are rigorous.

- The authors are careful in evaluating the strengths/weaknesses of their work. The work seems to be built upon an established literature and the review of related works are informative.

- Minor comments:
  - Definition 1.1: "and after inspecting which samples can be corrupted, it may change the label to an arbitrary value" --> I interpreted this as "change each label to an arbitrary value" or "change the labels to arbitrary values". Please clarify.
  - bit complexity of the parameters and samples: this is mentioned a few time but are not defined
  - the sentence starting at Line 140 is very difficult to read

**Originality**:

-  The paper consider a slightly weakened model (in comparison to the the contamination model), but leads to more generality in the underlying data distribution.

- The result of the work is significantly different (stronger) from the closest comparisons ([32] and [10]), and the methods are designed specifically to address the limitation of those works.

- The generalization of the approach from linear to ReLU regressions are non-trivial in both technical analysis and algorithmic designs.


**Significance**:

- As stated in the Originality section, the result of the work is significantly different (stronger) from the closest comparisons ([32] and [10]) and seems to extend existing works in a demonstrable way.

- On the other hand, I'm unfamiliar with some pieces of related work and don't have a strong conviction in how it will impact the field, and would leave the judgement up to other reviewers.


**Time Spent Reviewing:**

3

---

> ### Author Response · Authors · 2021-08-10
> **Author response for Reviewer Dpt6**
>
> We thank the reviewer for the appreciation of our paper and the helpful suggestions.
>
> For Definition 1.1, the adversary may change the labels of the (potentially) corrupted samples to arbitrary values. We will also update the paper with clear definitions and discussions with regard to the bit complexity of the parameters and samples.

---

### Official Review · Reviewer_vpXo · 2021-07-16

**Rating:** 6
**Confidence:** 3

**Summary:**

The aim of this work is to propose a computationally efficient algorithm for linear and ReLU regression in the presence of a bounded adversarial Massart noise model. It has been tried to make the assumptions on the underlying data distribution as mild as possible, not to mention that (as claimed by the author(s)) some assumptions are necessary from an information-theoretic point of view.

The core idea is to consider the unknown corrupted samples as outliers which do not follow the simple linear or ReLU rules between their features and corresponding labels. This should be added to the fact that (based on the paper's assumption) the fraction of corrupted samples is less than 1/2, so the clean data points have the majority. This way, author(s) have proposed a linear transformation on the samples in order the make them "Radially Isotropic", so (with high probability) the ratio of outliers to clean samples remain as low as possible in *every direction* of the space. Then, some existing robust estimators such as sum of $\ell_0$ or $\ell_1$ models have been shown to remove the effect of outliers and acquire the true parameters. I haven't completely checked the proofs, however, the overall idea makes sense to me and is, in fact, quite interesting. In any case, I am not completely familiar with this line of research so I wait to see other reviews in order to assess the novelty of techniques that have been utilized in this work.

With respect to weaknesses, paper lacks proper discussion at some points which I have explained in the "Main review" section. Also, author(s) have assumed a completely clean and noise-free model except for the adversarial part which still includes more than half of the samples. This assumption is a little worrisome in practice, where some minimum levels of, for example, additive noise are *always* present. How robust or sensitive is the proposed method and its theoretical guarantees when the presumed ideal noise-free environment is minimally perturbed?

Overall a well-motivated and fairly well-written paper with some interesting theoretical achievements (as far as I am aware). My vote at this stage is weak-accept.


**Limitations And Societal Impact:**

No problem here.

**Main Review:**


My main concerns are as follows:

Author(s) seem to have a good grasp of existing literature in this area, but it hasn't been completely reflected in the Introduction section. Some discussions and comparisons are still vague and need clarification. I believe the classification of prior advancements in terms of: achieved guarantees, distributional assumptions and etc. can be improved by some reorganization, which also helps the reader to position the current work w.r.t. others, more effectively.

The adversary described in Definition 1.1 does not match with the one that has been explained in Lines (117:119). In Def 1.1, adversary is allowed to alter (at will) each point $\left(x_i,f\left(x_i\right)\right)$, with probability $\eta\left(x_i\right)\leq\eta$. No information regarding $\eta\left(\cdot\right)$ has been given except that it is bounded by the constant $\eta$. That means the function $\eta\left(\cdot\right)$ can also be chosen by an adversary. That is different from the explanation given in Lines (117:119): "we consider a more restricted adversary that is presented with a uniformly random $\eta$ fraction of the points, which can be corrupted arbitrarily at will". So a natural question would be which of them is finally considered in this work?

Also, more discussion w.r.t. the adversarial perturbation model considered in this work would be helpful. It is not as strong as the "contamination model", but obviously hurt more than a purely random noise. What is the intuition behind this particular noise model?

In Section 1.2, the transition from ReLU and linear regression in the presence of Massart noise to $\ell_0$ or $\ell_1$-regression is not smooth. More explanations are needed here, otherwise the core idea behind the results become somehow ambiguous and hard to understand.


------------------------------------------------
Minor comments:

-(Line 107): What author(s) mean by "It remains an interesting open problem whether similar PAC learning guarantees can be obtained for the case of ReLU regression."? Aren't the results from Theorem 1.3 associated with the most general case? Or author(s) are referring to a more general adversarial noise model rather than Massart noise here?

-(Line 144): I guess it should be $w=A^Tw'$, right? It might not be important since $A$ is later assumed to be symmetric. However, it may confuse the readers.

-(Definition 1.4): What is $\mathcal{S}^{d-1}$? Is it the surface of $d$-dimensional unit sphere?

**Time Spent Reviewing:**

4 hours

---

> ### Author Response · Authors · 2021-08-10
> **Author response for Reviewer vpXo**
>
> We thank the reviewer for the helpful suggestions and comments. Below we address the specific questions from the reviewer.
>
> **Robustness under Model Misspecification:** It is indeed an interesting open question and a promising future direction to extend our work to cases where the true function is not actually a ReLU and allow for some form of model misspecification or additive noise as the reviewer suggests. Our theoretical results focus on the realizable model for learning ReLUs (and linear functions) with Massart noise. This is an important and well-studied setting even for the noiseless case (corresponding to $\eta = 0$), and has been the focus of a number of prior works, including recent papers appearing in top venues (see references [23, 31, 45, 49] and Lines 66-73).
>
> **Description and Definition of Adversary:** We define the Massart adversary in Definition 1.1 and use this definition throughout the paper. In fact, this definition is equivalent to the informal description given in Lines 117-119. This is because the adversary of Lines 117-119 does not have to change all the labels of the randomly selected $\eta$-fraction. By keeping some of the labels clean, this adversary is equivalent to that of Definition 1.1.
>
> **Intuition Behind the Massart Model:** As the reviewer has mentioned, the Massart noise model is stronger than purely random noise and weaker than the strong contamination model, which has computational hardness results even for well-behaved distributions, as described in Lines 32-34. It aims to capture cases where the samples that are (potentially) corrupted are not correlated, i.e., a uniformly random subset of examples is corrupted. The adversary may corrupt the values of these samples arbitrarily in correlated ways, but may not choose which points to corrupt in advance. This model allows us to escape the computational hardness results that apply when the adversary is all-too-powerful, and obtain efficient and robust regression algorithms under minimal assumptions about how the values are corrupted.
>
> **PAC Learning ReLUs in Line 107:** PAC learning ReLUs is the task of learning a hypothesis $h$ such that $\Pr_{x \sim \mathcal{D}_x}[h(x) \neq \mathrm{ReLU}(w^* \cdot x)] \leq \epsilon$ with probability at least $1-\delta$ *without* any distributional assumptions on $\mathcal{D}_x$. So it suffices to learn a hypothesis that outputs similar labels as $w^*$ without the hypothesis having to be exactly $w^*$. For instance, we can PAC learn linear functions even in the case that it is information-theoretically impossible to recover $w^*$ exactly, as shown in Appendix D. In the case of Theorem 1.3, we make a mild assumption that $\Pr(w\cdot x \geq 0) \geq \lambda$ to *exactly* recover $w^*$, while PAC learning would not make any assumptions and output a close hypothesis $h$.
>
> **Comment regarding Line 144:** Yes, we did not include the transpose since $A$ is symmetric, but we agree that this may be confusing. We will fix the typo and keep our notation consistent.
>
> **Definition of $\mathcal{S}^{d-1}$:** Yes, the reviewer is correct. We will add this definition for clarity.

---

> > ### Comment · Reviewer_vpXo · 2021-08-17
> > **Final comments to authors**
> >
> > Thanks for your response,
> >
> > Most of my concerns are obviated.
> > However, I keep my score since (at least IMO) the writing of the paper can still be greatly improved and the contents could become more accessible by the general audience. Moreover, some of the transitions between topics are not smooth enough.
> >
> > Other than that, this work seems to be a valuable technical contribution to the NeurIPS community.

---

### Official Review · Reviewer_463t · 2021-07-16

**Rating:** 7
**Confidence:** 3

**Summary:**

The authors study exact parameter recovery in ReLU regression under a generalization of the semi-random Massart noise model. Assuming certain distributional anti-concentration, the authors propose and analyze an algorithm that runs in polynomial time and recovers the exact parameters with high probability, given a sample of size polynomial in the input dimension and the (inverse of) the Massart parameter. Along the way, they also provide similar guarantees for the simpler case of linear regression.

**Limitations And Societal Impact:**

Yes

**Main Review:**

The paper is a joy to read. The study is important and relevant to the NeurIPS community. To the best of my knowledge, the results are novel and significant. I checked the proofs only at a high level, but the claims seem sound to me.

A couple of comments/questions:
1. The algorithm design as well as the proofs -- both in linear/ReLU regression -- leverage insights and techniques from the work of Hopkins et al. 2020. While this paper is cited in the appendix, there is no reference to it in the main text. I urge the authors to discuss this reference in the main text.
2. Given suitable anti-concentration assumptions, what are the main challenges in extending the results from linear regression to ReLU regression? The guarantees in Theorems 1.2 and 1.3 are very similar, and under the anti-concentration assumption for ReLU regression, proof of 1.3 seems like a straightforward extension of 1.2.


**Time Spent Reviewing:**

6

---

> ### Author Response · Authors · 2021-08-10
> **Author response for Reviewer 463t**
>
> We thank the reviewer for the appreciation of our paper and the helpful suggestions.
>
> Regarding the reviewer’s first comment, we note that our results in the main body of the submission are self-contained and do not depend on Hopkins et al.; only our extended results in the appendix require the facts proved in Hopkins et al. However, we agree that their work provides relevant insight and techniques. We will discuss their work when discussing radial isotropy in the Technical Overview section.
> Regarding the second comment, we note that ReLU regression is significantly more challenging than linear regression. A lot of recent literature aims to address the intricacies of ReLU regression stemming from the non-convexity of $\ell_1$ and $\ell_2$ regression. We have already highlighted these difficulties in Section 1.2 (Lines 156-174).
>
> At a high-level, if we knew a priori which points lie on the positive side of the ReLU, ReLU regression would indeed be a straightforward application of linear regression --  since we can learn the parameter vector $w$ with these points. In the presence of noise, however, it is unclear which region corresponds to the positive part and we must learn it simultaneously. This is one of the main challenges that we overcome. Our algorithm builds on the ideas developed in our linear regression warmup, to iteratively improve the guesses about the positive ReLU region yielding a separation oracle. We will incorporate this additional intuition in the revised version of the paper.

---

> > ### Comment · Reviewer_463t · 2021-09-02
> > **Post-rebuttal**
> >
> > Thanks for the feedback. I keep my initial score and recommend this paper for acceptance.

---

### Author Response · Authors · 2021-08-10
**Joint Response**

We thank the reviewers for their time and effort in providing feedback. We are encouraged by the positive feedback, and that the reviewers appreciated the paper for the following: (i) significant and/or interesting results (463t, vpXo, Dpt6), (ii) technical contribution (463t, Dpt6), and (iii) organization/clarity (vpXo, Dpt6, 8NsY). We address the individual questions and comments by the reviewers separately.

---

### Decision · Program_Chairs · 2021-09-27

**Decision:**

Accept (Poster)

**Comment:**

This paper considers the problem of ReLU regression under the Massart noise model that has recently been studied extensively for classification problems. The main result of the paper is an algorithm that does exact parameter learning under certain distributional assumptions.

All the reviewers appreciated the results of the paper. While it builds on prior techniques in the area, the technical novelty in the work is high enough. Certain technical questions raised by the reviewers were subsequently resolved by the authors' response. Overall this is a solid theory paper and I recommend acceptance.